# Development of a High-Accuracy Statistical Model to Identify the Key Parameter for Methane Adsorption in Metal-Organic Frameworks

**Kaushik Sivaramakrishnan *** and **Eyas Mahmoud**

Department of Chemical & Petroleum Engineering, United Arab Emirates University (UAEU),
Al-Ain 15551, United Arab Emirates
* Correspondence: kausiva@uaeu.ac.ae; Tel.: +971-58-964-9128

**Abstract:** The geometrical and topological features of metal-organic frameworks (MOFs) play an important role in determining their ability to capture and store methane ($CH_4$). Methane is a greenhouse gas that has been shown to be more dangerous in terms of contributing to global warming than carbon dioxide ($CO_2$), especially in the first 20 years of its release into the atmosphere. Its accelerated emission increases the rate of global temperature increase and needs to be addressed immediately. Adsorption processes have been shown to be effective and efficient in mitigating methane emissions from the atmosphere by providing an enormous surface area for methane storage. Among all the adsorbents, MOFs were shown to be the best adsorbents for methane adsorption due to their higher favorable steric interactions, the presence of binding sites such as open metal sites, and hydrophobic pockets. These features may not necessarily be present in carbonaceous materials and zeolites. Although many studies have suggested that the main reason for the increased storage efficiencies in terms of methane in the MOFs is the high surface area, there was some evidence in certain research works that methane storage performance, as measured by uptakes and deliveries in gravimetric and volumetric units, was higher for certain MOFs with a lower surface area. This prompted us to find out the most significant property of the MOF, whether it be material-based or pore-based, that has the maximum influence on methane uptake and delivery, using a comprehensive statistical approach that has not previously been employed in the methane storage literature. The approach in our study employed various chemometric techniques, including simple and multiple linear regression (SLR and MLR), combined with different types of multicollinearity diagnostics, partial correlations, standardized coefficients, and changes in regression coefficient estimates and their standard errors, applied to both the SLR and MLR models. The main advantages of this statistical approach are that it is quicker, provides a deeper insight into experimental data, and highlights a single, most important, parameter for MOF design and tuning that can predict and maximize the output storage and capture performance. The significance of our approach is that it was modeled purely based on experimental data, which will capture the real system, as opposed to the molecular simulations employed previously in the literature. Our model included data from ~80 MOFs and eight properties related to the material, pore, and thermodynamics (isosteric adsorption energy). Successful attempts to model the methane sorption process have previously been conducted using thermodynamic approaches and by developing adsorption performance indicators, but these are either too complex or time-consuming and their data covers fewer than 10 MOFs and a maximum of three MOF properties. By comparing the statistical metrics between the models, the most important and statistically significant property of the MOF was determined, which will be crucial when designing MOFs for use in storing and delivering methane.

**Keywords:** metal-organic frameworks; linear regression; multicollinearity diagnostics; linear and non-linear correlations; relative importance of explanatory variables; partial correlations; standardized regression coefficient estimates; variance proportions and distributions; pore surface area; material density; largest cavity diameter; isosteric adsorption energy; methane uptake and delivery

# 1. Introduction

## 1.1. Scheme Depicting the Main Content of This Work

Figure 1 below shows a circular scheme of the main content described in this work from introduction to conclusions topic-by-topic.

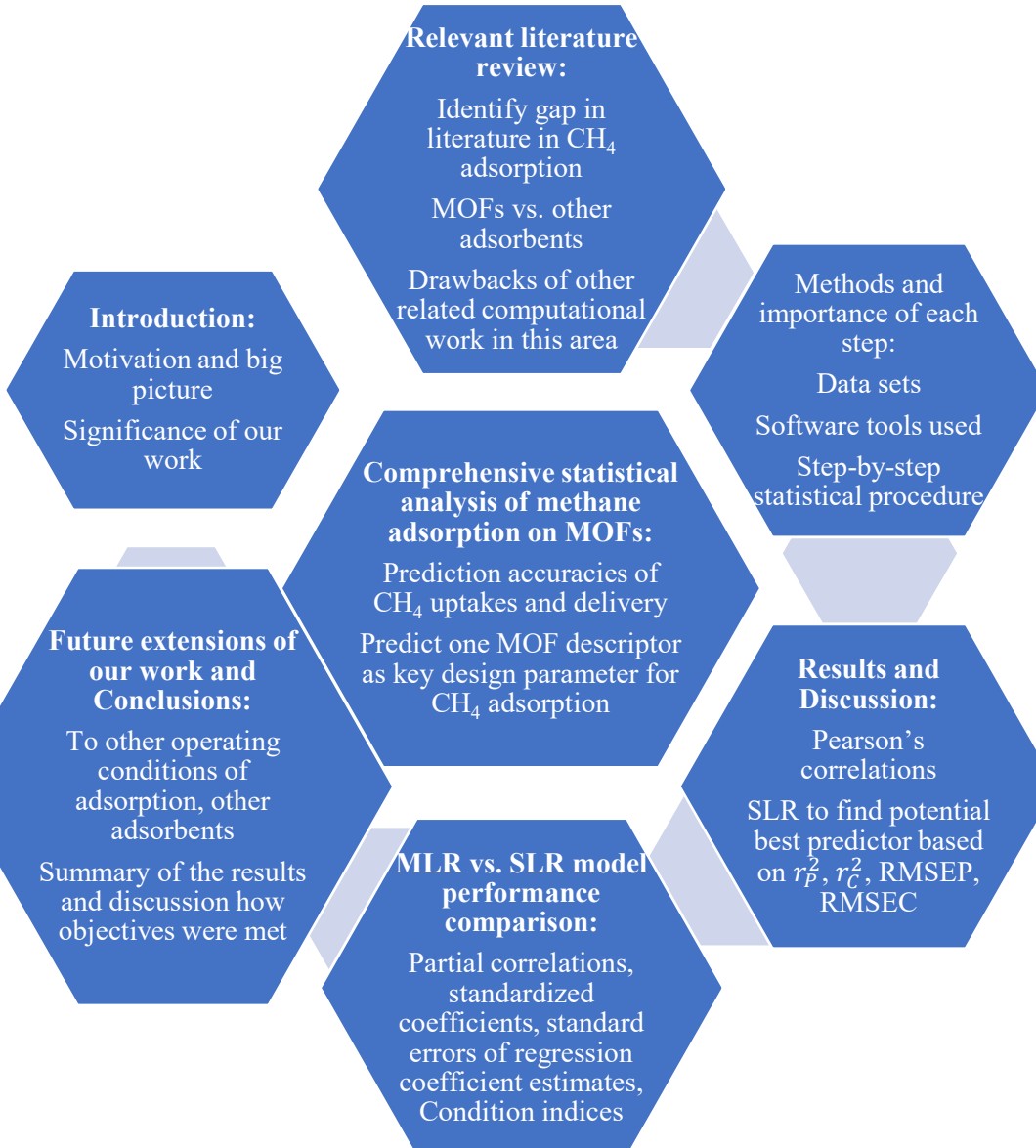

**Figure 1.** Schematic of the flow of the main content in this work.

## 1.2. Motivation and the Big Picture

Methane ($CH_4$) is a major constituent of natural gas that is used for heating homes and for generating electricity. Natural gas can be substituted for coal or oil, but they are more difficult to handle because of their heaviness and viscosity. Hence, methane has a very important place in modern times, but it can pose dangerous consequences, even when released into the atmosphere in the smallest of quantities. Global warming has been of serious concern for the past three to four decades but has currently reached unimaginable levels, with the regular occurrence of destructive hurricanes, heat spells, wildfires, and flash floods being attributed to the warming up of the earth's atmosphere. It has long been known that methane and carbon dioxide ($CO_2$) are the two major constituents of the greenhouse gases (GHGs) that contribute to global warming. The difference between the

gases is that methane is short-lived and is 30 times more potent than $CO_2$ in the first 20 years after its release (measured in terms of global warming potential (GWP)), while $CO_2$ remains in the environment for longer and is harmful in the long term [1]. The Environmental Defence Fund (EDF) determined that the concentration of methane in the atmosphere is increasing faster now than its rate of increase in the 1980s and that at least 25% of today's global warming is being driven by methane released into the atmosphere that is associated with human activity [2].

In the recently concluded United Nations Climate Change COP26 conference in 2021 in Glasgow, Scotland, many countries that make significant contributions to the oil and gas industry announced that they are working toward the goal of reaching net-zero greenhouse gas (GHG) emissions by 2050, with countries such as India targeting the year 2070 to achieve that objective [3]. Furthermore, the United States, Canada, and more than 80 other countries pledged to reduce their methane emissions by around 30% from their 2020 levels by the year 2030. In most cases, the actual emissions are higher than the estimates. For example, 13% of Canada's current GHG emissions can be attributed to methane, about 1.5 times higher than its estimates, and the primary contributors to methane emissions are agriculture and the oil and gas industry [3]. Interestingly, agriculture alone contributes 30% of the methane emissions via a process called enteric fermentation that occurs during the digestive cycle in cattle, producing $CH_4$, which is eventually released into the atmosphere. It has been shown that feeding the cattle additives such as sunflower oil and 3-nitrooxypropanol (3-NOP) to supplement their diet can significantly reduce their methane emissions [4]. A biodigester can also be effective in capturing the methane from farm waste and converting it into energy. The primary reason for methane production is the micro-organisms that act upon stored manure. This can be countered by altering the manure's acidity levels, which can then neutralize the micro-organisms. In this way, methane emissions from the agricultural sector can be reduced or harnessed.

About 13 million metric tons of methane were emitted by the oil and gas industry in the United States between 2012 and 2018 [5]. To put it in perspective, this was 60% more than the amount estimated by the Environmental Protection Agency (EPA) and this loss was worth approximately 2 billion USD (~8 billion AED). In addition, the Permian Basin, which is located in Texas and New Mexico and is the largest oil and gas field in the world, releases enough methane per year to fuel about 2 million homes. Simply put, the atmospheric presence of methane is responsible for one-third of the current global heating and, although $CH_4$ is more dangerous in the short term compared to $CO_2$, it can also be converted to $CO_2$ by atmospheric oxidation [6].

*1.3. Significance of Our Work and the Gap in the Literature*

The significance of reducing worldwide methane emissions is that a reduction of 30% of the current emissions can result in a 30% reduction of the global warming rate [7]. The above-mentioned data gives more than adequate motivation to tackle the issue of methane emissions. There are two ways this can be achieved: (i) reducing/mitigating emissions from the source or moving toward employing zero-carbon emission technologies; (ii) removing existing carbon from the atmosphere [8]. Both of these approaches are important in order to achieve the goal of a net-zero concentration of carbon in the atmosphere. In this work, we will be focusing on the second approach. To achieve this, we will store the methane at a pressure of around 35 bar and a temperature of 298 K. These values were used as the experimental conditions because in methane storage applications, as in those for natural gas-powered vehicles, the upper operating pressure used for storage is 35 bar and 298 K. In addition, the United States (US) Department of Energy (DOE) program for $CH_4$ storage systems, known as the "methane opportunities for vehicular energy (MOVE)" program has set the storage targets at a pressure of 3.5 MPa (35 bar) and 298 K. Changing the operating conditions will only majorly affect isosteric adsorption energies and the uptake and delivery of methane, while pore structural descriptors such as the volume, surface area, and density will remain the same. We are confident that the developed model in this work is versatile,

robust, and capable of handling the new data sets arising due to varying temperature and pressure conditions, as well as from synthesizing new metal-organic frameworks (MOFs).

Existing methods for this approach include storing carbon in agricultural lands and forests, while some technologies such as machines that suck carbon out of the atmosphere are at the testing stage. Fast-paced machines that mimic natural rock weathering are in the process of being built as well [8]. More than 10 gigatons of carbon need to be removed from the atmosphere every year, which is twice the amount of carbon emissions from the oil and gas industry.

Among the different methods to remove methane, such as absorption, membrane separation, and adsorption, adsorption has been shown to be most efficient and effective method [9]. The challenge in membrane separation is that the scaling up of the process increases the risks associated with higher pressure drops. The steric interactions between methane and an adsorbent are similar to that of nitrogen and the adsorbent, in part due to their similar molecular sizes. Hence, this increases the difficulty of separating methane from air mixtures [10]. Although some carbonaceous adsorbents have been shown to moderately meet the storage targets set by the US Department of Energy (DOE) for material-based $CH_4$ absorption, most zeolites, activated carbons, and inorganic materials fall below the target [11]. Moreover, they have a higher selectivity toward $CO_2$ due to their stronger interactions, which are fueled by higher polarizability due to its larger size [12,13]. This work focuses on using metal-organic frameworks (MOFs) as the adsorbent. MOFs have emerged in the past 20 years and work by capturing methane through physisorption. They are a relatively new class of micro- and meso-porous materials that consist of a central metal ion attached to organic ligands, and act like "crystal sponges". These were originally developed for hydrogen storage and $CO_2$ capture but have since been shown to meet the DOE targets for methane storage [14,15]. The main advantages of MOFs over other contemporary high-performance adsorbents used for $CH_4$ separation from gas streams, such as inorganic molecular sieves, are that they possess much greater surface areas, high porosities, and easily immobilized functional sites, which make them highly selective toward $CH_4$ [16]. The pore/cage size, shape, and other material properties can be tuned by changing the construct of the assembly (for example, the coordination preference of different metal ions can be altered) according to the desired functionality [17]. Ursueguia et al. [18] also reported that MOFs are a much easier and more efficient option than ventilation air methane (VAM) solutions, due to the latter's environmental safety risks associated with leaks.

With the amount of data available, there is an opportunity for a more rigorous statistical analysis in relating material properties with the storage performance of the MOFs. Previous work conducted by Mahmoud [19] established the basic regression correlations between separate MOF properties taken separately with the $CH_4$ storage performance metrics; the model adequacies were only measured using the coefficient of determination ($R^2$) [19]. Although this study, along with numerous other works in the literature, suggested that the MOF's surface area is a key design property, based on simple QSP correlations, there are other studies that provide evidence as to the importance of a different parameter that influences methane storage performance. A full summary of these studies and their relevance to our work is given in Section 2, the literature review. However, these previous works have only considered data from 10 MOFs or fewer and use a maximum of three input parameters. Thus, the value of the current work rests in the use of a more rigorous statistical treatment of the data by considering eight input variables spanning over 80 MOFs, focusing on the ability of a single variable to predict the storage performance independently. This is important since the input material properties are highly correlated among themselves, which reduces their individual influence on the output performance.

We will elaborate further on the correlations between the explanatory input variables in Section 3. Illustratively, in certain studies that suspected the lower importance of the surface area, it was seen that certain MOFs, although having twice the surface area, showed a lower storage capacity. Moreover, although many research works have determined only

the qualitative relationships between the functional properties of MOFs, such as pore size, surface area, density, and adsorption entropy and the methane storage performance, a rigorous statistical approach is lacking [15]. Therefore, the need arises for us to find a key design parameter that influences the storage performance, using a simpler, less time-consuming but more comprehensive approach. The current work uses chemometrics and statistical analysis to provide the answer. The significance of this approach is that it provides fewer parameters for the faster design of new MOFs or for the modification of the current design by tuning only the single most important material property to achieve optimal performance. The types of parameters, including the material, pore, and thermodynamic properties, that describe both the geometrical and topological features of the MOFs have an influence on the storage performance and are described in the following section.

## 2. Relevant Literature Background, Specific Aims, and Key Contributions of This Work

The nature of this work demands that we provide an extensive report of the latest methane detection and storage techniques used in the industry, as well as the research conducted into methane storage applications using MOFs involving computational and mathematical approaches. In doing so, we wish to mention what particular value the addition of our statistical methods will bring to improving the storage performance of $CH_4$. This section will review the different methane detection technologies, why MOFs are more suited to methane than $CO_2$, the relevant studies that formed the basis of our hypothesis and objective, and a few of the mathematical approaches used in quantifying methane storage performance in MOFs, and also highlight the advantages of and reasons as to why our work will fill the gap in the literature by significantly contributing to this area.

### 2.1. Methane Detection Technologies

The current increase in the concentration of methane in the atmosphere has been attributed by the scientific community to the following factors: (i) the natural mechanisms causing methane breakdown have been weaker; (ii) agriculture and wastewater treatment provide biogenic sources, while wetlands and flood zones provide natural sources; (iii) seeps and mud volcanoes provide natural geological sources [20]. The study by Hmiel et al. [20] indicates that the anthropogenic sources of methane, such as fossil fuel extraction, are currently underestimated and must clearly be reduced. The current methods to mitigate these emissions have evolved over the last few years, of which methane detection is the initial step. This step is difficult because methane is an odorless and colorless gas. Satellites have been developed and commissioned that are highly sensitive to methane gas, such as MethaneSAT [21]. In general, two types of approaches are currently used for methane tracking: bottom-up, and top-down approaches [5]. The bottom-up approach takes the measurement at the source, while top-down approaches, such as employing methane sensors on an aircraft hovering over a particular area, have a more holistic impact since they bring a larger area into consideration. Hence, these two approaches should be used in tandem. However, most of these methods are quite expensive and complex; hence, the appropriate technique should be chosen while keeping the economics in mind.

### 2.2. Comparison of MOFs vs. Inorganic Adsorbents and Suitability of MOFs in Terms of $CH_4$

Once the methane is detected, choosing the right technology, equipment, and materials are important for its efficient removal and storage. As highlighted in the previous section, adsorption is the preferred process but evaluating which adsorbent material to use for methane can be difficult. Since storing methane in the form of compressed natural gas (CNG) has the disadvantage of high-pressure containment requirements, adsorbed natural gas (ANG) has been seen as an alternative means of capture using a porous adsorbent [22]. Storage space becomes critical for methane due to its low volumetric energy density, which is another reason why the scientific world has moved toward adsorbents. Furthermore, Gholipour and Mofarahi [23] stated that adsorption selectivity is heavily influenced by

steric effects, the differences in size of the adsorbate and the pores in the adsorbents, the surface charges, and the polarizability of the adsorbate molecules. For the purpose of comparing the absorption capabilities of different adsorbents, Kim et al. [24] conducted adsorption experiments with over 190 samples, creating more than 87,000 simulations for the crystallographic structures of inorganic adsorbents, and found that materials with a higher quadruple moment, such as $CO_2$, have a higher affinity to inorganic adsorbents; when using carbonaceous materials such as activated carbons/charcoals or carbon black, the higher polarizability of $CO_2$ makes the adsorbent material more selective toward $CO_2$ rather than to $CH_4$.

It can be recognized from the literature on adsorption regarding methane that MOFs offer the best solution as they have a high ability to store and deliver $CH_4$, due to their favorable geometrical and topological features. These features ultimately decide the input parameters for our statistical model; these are density, pore volume, BET and accessible surface area, void fraction, and largest cavity diameter (LCD), which are all key to the design and synthesis of MOFs. Kondo et al. [25] published one of the first research works that explored the use of MOFs for storing methane through adsorption. However, as in most of the later studies, this work presented only the data for the surface areas and porosities of the MOFs and the corresponding methane uptakes. Ma et al. [26] further showed that this uptake of methane can exceed the US DOE target in MOFs. An extensive review article by Okoro et al. [17] suggested that methane gas uptake calculation for MOFs was based on crystallographic density, while Senkovska and Kaskel [27] reported in 2008 that this technique needs more work to determine the storage capabilities of MOFs from the crystal density alone.

### 2.3. Studies Relevant to Our Hypothesis and Objective

Experimental data for the uptakes of IRMOF-6 and MOF-5 as methane adsorbents with fully resolved crystal densities showed significant performance potential, which increases the data availability for statistical models [28]. In the early stages of MOF usage, the methane storage capabilities of well-known MOFs, such as HKUST-1 (copper-based, with benzene tricarboxylate as the ligand) and MIL-53-Cr (chromium-based dicarboxylate), which have been used by several research groups, yielded results that were quite close to the DOE target (in the year 2000) of 180 $cm^3/cm^3$ volumetric uptake units (calculated at STP), in conditions of 298 K and 35 bar [29]. This target was increased to 350 $cm^3/cm^3$ in 2012. Up until 2010, the Cu-based MOF, PCN-14, had set the record by showing a maximum methane storage capacity of 220 $cm^3/cm^3$, as reported by Ma et al. [30] under the same conditions. In 2008 and 2009, other research groups reported values of up to 190 $cm^3/cm^3$ for methane uptake, thus clearing the DOE target comfortably. An interesting observation in these data was that the MOF-5 used by Zhou et al. [31], with a higher surface area, exhibited half the adsorption capacity than that of PCN-14 (as employed by Ma et al. [30]), which actually cleared the previous DOE (2000) cut-off but had a lower surface area. Furthermore, the density of PCN-14 was higher than that of MOF-5, indicating the possibility that density might be a better predictor of storage capacity than surface area, contrary to the general notion that pore surface area correlated best with storage capacity. Our results in the present work corroborate this observation in Table 1 taken from the review article by Zhou [15], and clearly warrants the need to apply statistical approaches when critically analyzing the MOF property-methane uptake data for trends and correlations and identifying the best predictors for MOF design by employing multicollinearity diagnostics. One pertinent question is: are there other parameters that play an important role in methane storage performance in MOFs? Is there statistical evidence for the observation that properties such as density can influence the ability to store and deliver methane better than surface area? If we can establish the answers to these questions, the design of MOFs can be appropriately modified to improve their storage capabilities.

Hence, we decided that the hypothesis behind our work would be that crystal density is the key parameter for determining the storage power of the MOFs; we were encouraged

by the preliminary results of our project, obtained from simple linear regression (SLR) models, wherein we found that density was the most accurate predictor of methane uptake and delivery in gravimetric units (Section 4.2). However, it was still to be seen if that quality individually contributes to output variance and prediction, which was established by multiple regression models and by including other contributing input variables (MOF material properties), such as the surface area, largest cavity diameter, etc., and analyzing their performance with stricter statistical tests and multicollinearity diagnostics to tackle inter-dependencies between the material, pore, and thermodynamic properties of the MOFs. This can also overcome the issues of overestimating storage performance when converting gravimetric uptake to volumetric uptake (for which an accurate measurement of crystallographic density is required) in order to compare the results with the DOE target. The developed statistical model in our project can directly estimate the uptakes from new density data. To the best of our knowledge, our rigorous statistical analysis will be the only one to encompass 8 different MOF material properties from a huge volume of experimental data taken as inputs, utilize multiple linear regression to validate input-output dependencies, and apply multiple multicollinearity diagnostics to isolate the parameter that is of greatest importance in creating the MOF design-methane storage/delivery performance output relationships.

The overall objective of this study is to identify the most important design parameter, arising from geographical or topological features of the MOF, that is directly responsible for determining their storage and influencing the deliverable capacity, which is applicable to both methane storage and delivery processes. One important application of our work is that this will help in making the design process easier and quicker, to create MOFs that can be used in methane storage tanks in automobiles around the world and in fuels for recreational activities such as barbecues and in the food industry. We will be using a comprehensive and rigorous statistical approach for achieving this goal since it is rapid, easy to implement, and reliable. Our experimental data covers 83 actual MOFs and 8 input properties, as compared to studies covering < 10 real MOFs and a maximum of 3 input properties, as seen in other works in the literature.

### 2.4. Evidence of the Importance of Thermodynamic Factors in Storage Performance

Since our proposed project aims to establish a statistical relationship between the material properties used for MOFs based on geometrical and topological features, with the performance metrics for methane storage and capture, we have reviewed some of the most important research that has previously been conducted in this area and have highlighted a gap in the literature, to show how our project is necessary and works to address this gap. Pore surface area was seen to have a strong correlation with methane uptake for a number of MOFs, as reported by Duren et al. [32] in 2008. In a couple of their papers, Wu et al. [33,34] utilized a combination of experimental characterization (neutron diffraction) and computational techniques to investigate $CH_4$ adsorption in several MOFs and found that certain open metal sites, such as unsaturated metal ions (through a Coulomb interaction), enhanced hydrophobic pockets that have a similar dimension to methane (3.8 Å) and interacted through dispersive forces, having a stronger affinity with $CH_4$ molecules. These parameters can also be characterized by the particular heat of adsorption ($Q_{st}$), which has been identified as a critical parameter that influences methane capture for mitigating emissions, as reported in our co-author's previous work [35]. These works further suggest that while pore surface area and volume are still important factors, the contribution of other topological factors seen in the MOF should not be ignored when working with adsorption pressures of 35 bar.

### 2.5. Key Contributions of Important Research Works Involving Computational, Parameter Estimations, Thermodynamic Approaches, and Their Drawbacks

It is important to note that identifying the mechanism of adsorption of methane is not the focus of our project, but it may potentially be a by-product of our work, since we

can try to corroborate the dominant mechanism for the binding of methane molecules to the MOFs by comparing it to the most important property for methane uptake that the statistical model shows. On that note, Wu et al. [33] recommended that increasing the open metal sites and volume percentage of the accessible small cages and channels, while minimizing the fraction of larger pores, can increase methane uptakes significantly. This process can be seen as a decrease in the crystal density, which was seen to have a negative correlation with methane uptake from our preliminary results and also to be the predictor with the highest accuracy (Sections 4.1 and 4.2), thus encouraging us to proceed with further analysis to isolate the density as being the primary predictor having a sole influence on the output. Molecular simulations such as those performed by Wu et al. [34] and, more recently, Sezginel et al. [36] can help to visualize the site structure; however, the drawback is that these sites can be dynamic and change over time, due to their complexity and heterogeneity, and change from experiment to experiment. Furthermore, Hechinger et al. [37] noted that the three-dimensional structural descriptors, which form the inputs to a quantitative structure–property relationship (QSPR) model, are generated from a generalized and most stable conformation of the MOF structure in molecular simulations; this might lead to erroneous results from the QSPR due to the dynamic nature of the MOF structure in real-time. It is therefore most logical to use experimental data to develop statistical models by which to investigate the most important parameter for methane storage and capture, and it is important to note that our entire work will be based on experimental data only. If the methane molecule interacts with several surfaces simultaneously, this brings the heat of adsorption into play, which we believe will be the determining parameter for methane capture to mitigate emissions [35].

With respect to the current DOE (2012) gravimetric target of 0.5 g/g, the methane uptakes of six promising MOFs (PCN-14, UTSA-20, HKUST-1, Ni-MOF-74, NU-111, and NU-125) were determined to meet the target. They were also seen to vary linearly with their pore surface area and pore volume and were inversely proportional to density in the research published by Peng et al. [38]. However, their work does not analyze which of these MOF properties predicted the gravimetric uptake most accurately, but the study reports that the packing efficiency of the MOFs needed to be taken into consideration, so that the MOFs are built to withstand higher mechanical pressures. It is important to note that packing densities are different from crystal densities, although the two are highly correlated. In 2016, Li et al. [39] utilized computational screening techniques to study thousands of MOFs and arrived at the conclusion that there was a physical limitation on methane storage of 200 $cm^3/cm^3$, which is below the DOE target at high pressures. An enormous resource of methane gravimetric uptake data for various MOFs at different pressures, ranging from 1 bar to 200 bar, has been reported in the recent literature [27,40–49]. The information given on the physical properties of the MOFs, such as specific surface areas and the presence of open metal sites, are also given in the respective works. The results from these works provide a great source of input and output data for the development of statistical models similar to those in our work, as they align with our objective of working with experimental data for our mathematical models.

However, most of these works involve quantitative analysis techniques that fall under the molecular simulations category and differ from what we intended to achieve. In 2017, Becker et al. [48] highlighted the importance of polarization by comparing the proposed polarizable force fields with orbital interaction energies obtained from density functional theory (DFT) calculations. The different heats of adsorptions were also calculated for $CO_2$ and $CH_4$ on MOFs with ten different metal ions, and these compared well with the experimental data. Force fields describe the interaction between adsorbents and adsorbates, whereas computational simulation tools employ quantum mechanics and compute adsorption isotherms for MOF-methane systems. Another recent work that computes adsorption isotherms, from which the adsorption energies can easily be derived, was published by Vandenbrande et al. [44] in 2017, wherein they compare several other force fields, computed through simulations, from the literature for methane adsorption in Zr-based MOFs. They

also point out that the major issue with quantum simulations is that there are significant quantitative differences in uptakes, which are predicted from the computed adsorption energies across systems under different experimental conditions since the predictions are highly sensitive to the computed potential energy surface (PES). The additional value of the current work comes in at this point, whereby our model will directly predict the uptakes with better accuracy, as indicated by our preliminary results, and will minimize the computational effort and hours involved in molecular simulations.

Interestingly, an adsorbent performance indicator (API) was developed by Wiersum et al. [43] in 2013; it considered only the adsorption energies and working capacities as the inputs but tried to address the performance sensitivity to process conditions by including weighting factors. This expression is dependent on pressure and compares very well with the experimental isotherms but does not consider other key properties in the model, such as the density and pore volumes. Methane sorption in five different MOFs was modeled after collecting the corresponding experimental data by Tahmooresi and Sabzi [47] in 2014, using them to calculate methane uptakes in various MOFs using the perturbed hard sphere chain (PHSC) equation of state, which included molecular dimensions and interaction energies as the input parameters. This thermodynamic approach also compared well with the experimental data from isotherms, but the study considers only five MOFs; the molecular parameters were also estimated using a group contribution method and might not represent the actual system properties at all times. Moreover, thermodynamic approaches tend to be both complex and time-consuming. High-performance MOFs have also been identified by the computational screening of 204 hypothetical MOF structures, with Zr as the central metal and OH and $CO_2$ as the ligands, as reported by Gomez-Gualdron et al. [50]. However, this work required the exact structures of a number of other MOFs as inputs, which are difficult to obtain in real time.

All the research works that have been mentioned in this section are summarized in the following table (Table 1), where brief descriptions of their main agendas and the studies' drawbacks are provided.

**Table 1.** Main agendas and drawbacks of the most important research works involving various computational techniques in methane adsorption.

| Authors | Description/Agenda | Parameters Considered | Drawbacks/Motivation for Our Work |
|---|---|---|---|
| Wu et al. [33] | Recommended minimization of larger pores and an increase in the percentage of small cages and open metal sites | Pore volume, accessible surface area, density | Purely experimental study and takes longer. Correlates with our results that density can be a crucial parameter |
| Wu et al. [34] Sezginel et al. [36] | Molecular simulations to visualize the site structures | Assumptions on pore size, volume, density to get adsorption energies | Does not account for dynamic changes in structure during experiments |
| Hechinger et al. [37] | DFT simulations provided 3-D structural descriptors that were taken as inputs to QSPR models | Pore size, pore volume, surface area | Assumption of most stable conformation can lead to erroneous energies |
| Peng et al. [38] | Methane uptakes of 6 promising MOFs were determined to meet the DOE 2012 target and also seen to vary linearly with their pore surface area and pore volume but inversely proportional to density | Pore surface area, pore volume, packing density | The analysis was not comprehensive enough to find out the best descriptor of the MOF that can predict the uptakes with the highest accuracy |

**Table 1.** *Cont.*

| Authors | Description/Agenda | Parameters Considered | Drawbacks/Motivation for Our Work |
|---|---|---|---|
| Li et al. [39] (2016) | The computational screening of MOFs -> revealed physical limitation to storage | Structural descriptors | Thermodynamic descriptors not considered Molecular simulations consume a great deal of time |
| Becker et al. [48] (2017) | Compared polarizable force fields vs. orbital interaction energies DFT -> adsorption isotherms -> adsorption energies Calculated $\Delta H_{ads}$ for $CO_2$ and $CH_4$ on MOFs with 10 different metal ions | $\Delta H_{ads}$, assumptions of structural descriptors such as pore size, volume, density | Computational tools involving quantum mechanics consume lots of time and computer power and memory |
| Vandenbrande et al. [44] (2017) | Computed force fields for Zr-based MOFs adsorbing $CH_4$ | General stable form of MOF assumed | Significant quantitative differences in methane uptakes, predictions are highly sensitive to the computed PES |
| Wiersum et al. [43] (2013) | Developed API considering weighting factors for performance sensitivity towards temperature and pressure | Adsorption energies, working capacities as inputs | Other material and pore properties of the MOF not considered whereas our model considers 8 input parameters |
| Tahmooresi and Sabzi [47] (2014) | Methane sorption in five different MOFs was modeled from experimental data PHSC) equation of state used—thermodynamic approach | Molecular dimensions and interaction energies used as inputs | Only 5 MOFs were considered, whereas our model considers 83 MOFs Other material and pore properties of the MOF were not considered Molecular parameters were estimated using a group contribution method—may not represent a real system |
| Gomez-Gualdron et al. [50] | Identified high-performance MOFs with Zr as central metal and OH and $CO_2$ as ligands from a computational screening of 204 MOFs | | Requires exact structures of MOFs as inputs—very difficult and time-consuming in the real system The MOFs used in the screening were hypothetical |

*2.6. Key Contributions of This Work*

In comparison to all these methods, our study offers an uncomplicated, simple but rigorous, and comprehensive step-by-step statistical approach to predicting methane uptake during storage and capture, and also identifying the key parameter for MOF design from experimental data on 83 MOFs. Some of the clear disadvantages of the methods adopted in these existing studies are: (i) the molecular simulations are not dynamic, as they consider not more than ten simulated MOFs and predict the performance using the resulting adsorption energies and potential energy surfaces (PES) obtained, which might not reflect the real system; (ii) the thermodynamic models require the estimation of molecular parameters, are complex and time-consuming, and consider fewer than three input parameters for five MOFs; (iii) the performance indicator expressions carry human-input weighting factors that may not be indicative of the real system and also consider only two input parameters and around ten MOFs. Overall, we have identified that one major gap in the methane storage and capture literature using MOFs is the lack of a rigorous statistical model that takes all the possible MOF material properties into consideration

and arrives at a single and most important parameter that can be used for MOF design, in order to efficiently capture $CH_4$ and maximize its storage and deliverable capacity. The advantage of developing such a model is that it will predict the target storage performance measures of the MOF with high accuracy.

We aim to isolate this key property that best predicts the output MOF performance, a property that simultaneously has an independent effect on performance, even in the presence of other correlated properties. This was achieved through extensive and thorough statistical techniques that were less computationally complex than simulations and consumed much less time. Although the methods we use are detailed fully in the Methods section (Section 3), here, we provide a brief summary of our approach.

Since it will be difficult to provide the names of all 83 MOFs and information about their structural, pore, and material descriptors and the thermodynamic properties used in this work, we will provide the names of some of the primary MOFs used here. Examples of some of the MOFs include HKUST-1, NiMOF-74, PCN-14, CoMOF-74, MgMOF-74, NOTT-109, PCN-11, NU-111, NOTT-100, NOTT-107, UTSA-20, Cu-TDPAT, PCN-68, $Zn_2(bdc)_2(dabco)$, MIL-53(Cu), and others. It can be seen that the MOFs used in this work possess different central metals attached to different ligands, which makes our model more flexible and applicable to real-time adsorbents for methane storage by increasing the changes of higher prediction accuracy for new data sets. This is especially applicable when new MOFs are synthesized. More detailed information about their structure (especially of the nanocages in NU-111, Ni-MOF-74, PCN-14, and UTSA-20) is provided in Figure 1 of the previous work by our co-author, Mahmoud [19]. The experimental data for the different MOF parameters and uptakes used in this work are already provided in the Supplementary Materials document of the study by Mahmoud [19].

After splitting the 83 observations into the calibration and validation data sets, we applied bivariate Pearson's correlations and simple linear regression (SLR) to the input and output variables on the calibration set to investigate the relationships between them, which will help us to find the best predictor for the outputs depicting methane storage performance. In the SLR models, we also investigated the different types of nonlinear fits, including quadratic, cubic, and exponential fits. Then, the best fit of type for each input variable was also evaluated on the validation set since the ability to predict the outputs on new MOF property data was one of the goals of this work. This will also render the model more adaptable and will mitigate any overfitting tendencies. Next, we applied multiple linear regression to tackle the phenomenon of multicollinearity that exists among the input variables, as evidenced by the high Pearson's correlations between them. The most important variable was identified by developing the MLR models, keeping the best predictor from the SLR models and adding one input variable at a time.

This way, the influence of the inputs on the output variables can be separated by looking for drastic changes in regression coefficient estimates, their standard errors, significances of the terms through $t$-statistics, evaluating the standardized coefficients, partial correlations, and other important multicollinearity diagnostics, such as eigenvalues, condition indices (CI), and variance proportion distributions, among the different input dimensions. These statistical metrics in the MLR modes are compared with those from the SLR models to isolate the most important material or pore property of the MOF, which would show minimal changes to these statistical measures. This is very important for quick and efficient MOF design and $CH_4$ storage for vehicular use, as natural gas represents a cleaner and more economical fuel than gasoline. The other goal of our study was to answer the important design question of whether the developed models can be applicable to new data for the identified MOF property of significance. Furthermore, we ensured the statistical significance of our models. Details of the methods employed in this work are described in Section 3.

### 3. Methods and the Significance/Importance of Each Step to Our Objective

*3.1. Data Sets and Splitting*

The data consists of eight properties related to the material, pore, and thermodynamics properties, which are as follows: pore volume (V$_p$), BET surface area, density, accessible surface area (ASA), accessible volume (AV), largest cavity diameter (LCD), pore-limiting diameter (PLD), and isosteric heat of adsorption (Q$_{st}$), spanning 83 MOFs and serving as the inputs in our regression models. The outputs are the measures of storage performance in the MOFs, known as uptakes and deliveries, taken in gravimetric and volumetric units. These are known as gravimetric uptake (GU), volumetric uptake (VU), gravimetric delivery (GD), and volumetric delivery (VD). The deliverable capacity is the amount of gas released between the upper storage pressure and 5 bar, required at the engine inlet of an adsorption-desorption process [19]. All the aforementioned input properties and output performance metrics are obtained using the experimental data provided in the study by Mahmoud [19]. The data is split into the calibration and validation sets, among which the correlations and regression models are developed using the former data set and are tested for real-time prediction accuracy and model adequacy on the latter data set. The number of observations in the calibration set varied between 23 and 51, depending on the input-output combination, due to some missing data for the material, thermodynamic, and pore properties, such as ASA and AV; the remaining observations from the 83 points were used for the validation set.

*3.2. Software Tools Used*

All the statistical analyses in this work were conducted using IBM SPSS Statistics (Version 28) and the software was run on a Windows 10 operating system.

*3.3. Steps in the Statistical Approach Followed in This Work*

All the correlations and regression models in this work are based on sample data since the population data are not known for this system. Here, the population refers to the set of eight-input material, thermodynamic, and pore properties taken for all possible existing MOFs, developed by measuring the gravimetric and volumetric uptakes and deliveries for these conditions at 298 K and 35 bar. We chose to work with a sample of 83 MOFs for which we had the experimental data. All the regression coefficient estimates obtained from the developed regression models can be considered to be an approximation of the true population data since the entirety of the population data was not considered in this work.

3.3.1. Pearson's Correlation and Its Significance

The first step involves determining the bivariate Pearson's correlations among the different input variables themselves, taken two at a time, as well as between the inputs and outputs, also taken two at a time [49]. The correlation is defined as follows:

$$r_{X-y} = \sum_{i=1}^{n} \frac{(x_i - x_{mean})(y_i - y_{mean})}{\sqrt{\sum (x_i - x_{mean})^2 \sum (y_i - x_{mean})^2}} \tag{1}$$

where $x_i$ and $y_i$ are the observations of the independent and dependent variable vectors, $X$ and $y$, respectively, and $n$ is the total number of observations in $X$ and $y$. Furthermore, $x_{mean}$ and $y_{mean}$ are the average of the observations of $X$ and $y$, respectively. The square of the bivariate correlations gives the coefficient of determination ($r^2$) for the SLR model between the dependent and independent variables. $r^2$ can also be interpreted as the proportion of shared variance between $X$ and $y$, for which the SLR model is being developed.

From the correlations between the inputs and the outputs, we estimated the multi-collinearity among the input variables and identified which input parameters of the MOF have a significant relationship with the output by analyzing the strength of their linear relationship. Having said this, it is important to note that we cannot directly conclude that the input parameter that has the highest correlation with the outputs is the best predictor

for that particular output because the interdependencies among the input variables cause a redundancy in the shared variance with the output. For example, if density has the highest correlation with one of the output variables, we need to further evaluate the correlations between density and other explanatory variables and check whether they are high and significant at the 95% confidence level (>0.8) [51]. Pearson's correlations are the first step to identifying the multicollinearity phenomenon among the input parameters that has been known to affect the regression model performance most significantly [52,53]. The significance of a correlation is also tested by applying two-tailed *t*-tests. This assumes that the data follows a bivariate normal distribution, which is confirmed by the plot between the cumulative distribution function of the observed function and the expected distribution for each variable [53]. A correlation is considered to be significant if the *p*-value in hypothesis testing is <0.05, which also means that there is only a 5% chance that the population correlation is equal to 0.

### 3.3.2. SLR and MLR Models—Assumptions, Regression Coefficient Estimates, and Standard Errors

The next step is to find the best fit for each input parameter with the output storage performance variables. Each input parameter is regressed with each output separately, using a simple linear regression (SLR) technique, and the fits of linear, quadratic, cubic, and exponential types are tested for the calibration data set. This will indicate which form of the variable has the potential to best predict the output variable since most data in real time are nonlinear. The basic relationship is a linear regression model of the form ($Y = b_1 * X_1 + b_0$), where $X_1$ is the input variable under a linear fit, and b0 and b1 are the intercept term and the unstandardized regression coefficient estimate. A classical assumption in regression analysis is that the errors or residuals of the model estimates need to be normally distributed; this can be verified from the residual plots. Another assumption is regarding the equal variance for the residuals, which necessitates that the mean of the residuals should be very close to 0, as is also verified in our work. The error term in the final equation of the linear regression model is given by N (0, $\sigma^2$), where 0 is the mean and $\sigma^2$ is the variance. There was also minimum variation in the measurement of the material, thermodynamic, and pore properties of the MOFs, which is also an essential requirement for regression analysis. The final requirement of regression analysis is that each experimental observation resulting in methane uptake needs to be independent, which is the case in our study. In other words, the measured methane uptake for an MOF with a set of geometrical and topological features is not influenced by another MOF.

The other transformed forms are modeled as follows: Quadratic: $Y = b_1 * X_1^2 + b_0$; Cubic: $Y = b_1 * X_1^3 + b_0$; Exponential: $Y = b_0 \exp(b_1 * X_1)$. In the case of quadratic and cubic fits being the best fits, the intercept term is ignored since it was found that the coefficient of the quadratic or cubic term was of the highest significance, as measured by the *p*-value of the *t*-distribution. Similar to the Pearson correlations, the *t*-statistic for calculating the significance of each regression coefficient estimate ($b_i$'s) is determined as:

$$t_{b_i} = \frac{b_i}{SE_{b_i}} \qquad (2)$$

where $SE_{b_i}$ is the standard error of the coefficient estimate $b_i$, and *i* is the number of the input variable.

Hence, it can be seen that if the standard error increases, the *t*-statistic decreases, as does the significance of that coefficient estimate. We call this term the coefficient estimate because it is applied to the sample data and not to the population data. The standard errors of the coefficient estimate can show higher values under 3 possible conditions: (i) the residuals or errors in the predicted values from the model increase; (ii) the correlations between the input variables increase; (iii) the variance of the predictors increases. In this work, we have taken 2 input variables at a time in the MLR models; hence, the standard errors will be calculated for each of the 2 regression coefficient estimates. As described

in Section 3.3.4, these input variables will be the best fits obtained from the SLR models, in order to confirm the relative importance of each variable in predicting the output. It is important to note that the number of degrees of freedom of *t*-distribution is $n - k - 1$, where $k$ is the number of explanatory variables and $n$ is the number of observations in the data set. The significance of each regression coefficient estimate is found by the area under the *t*-curve and is given directly using an IBM SPSS software package. It is also interesting to note that when the model is standardized, the intercept term disappears; the standardized coefficient obtained is the same as the bivariate Pearson's correlation between the input and the output variables. This is applicable only for the SLR model since in MLR, the influence of both input variables comes into play.

### 3.3.3. Model Performance Evaluations and the Significance of the Overall Model

Once the SLR and MLR models were developed on the calibration set with each of the 4 outputs and 8 input properties of the MOFs separately, the predictions of the output variables from the model were evaluated against the experimental data available for both the calibration validation sets, using 2 measurement metrics: (i) coefficient of determination ($r_C^2$ and $r_P^2$ for the calibration and validation set, respectively) for the SLR and $R_P^2$ and $R_C^2$ for the validation and calibration set, respectively, in the MLR models; (ii) the root mean square error (RMSEC and RMSEP, evaluated on the calibration and validation set, respectively) as model-adequacy parameters. The RMSEC is very similar to the residual standard deviation (RSD), which is evaluated according to the output predictions of the SLRs and the MLRs to test the model performance. A lower value of RSD/RMSEC and a higher value of $R_C^2$ mean that the prediction accuracy is high on the calibration set. However, the best predictors for each output are primarily chosen based on the highest $r_P^2$ and $R_P^2$ and RMSEP since the performance on new data sets is more important than performance on the same data set on which the model was developed. Furthermore, the difference between the RMSEC and RMSEP will give us an idea of the overfitting tendency of the model; a lower value of RMSEP–RMSEC would indicate that the outputs are predicted well for any general data set using the corresponding predictor variable. Otherwise, it becomes a case of good fitting only with respect to the calibration data set; the model will perform badly for any new input data regarding MOF material, thermodynamic, and pore parameters. This is also considered to show the high flexibility of the model, which is also not a good thing for model performance.

All the regression coefficients, their standard errors, *t*-statistics, *p*-values, and the model performance measures of RMSE, $r^2$ and $R^2$, are provided directly by IBM SPSS. All the values were calculated and verified by hand as well, except the *p*-values, which are directly provided by IBM SPSS since the area under the curve is complicated to calculate. It is very interesting to note that the coefficient of determination in an SLR model is the square of the bivariate Pearson's correlation between the input and the output variables. This is not necessarily true for the MLR models because the variance in the output is shared by more than one variable. Another way in which $R^2$ is calculated and verified by hand in our work is by determining the bivariate correlation between the model-predicted value and the experimental value for that output variable corresponding to the observation.

The *F*-statistic is another metric that is used to test the significance of the analysis of variance (ANOVA) or, in other words, the overall $R^2$ (for MLR) or $r^2$ (for SLR) of the model. It is evaluated by dividing the mean squared model (MSM) by the mean squared error (MSE). The MSM is calculated by subtracting each of the predicted values for the observations from the mean of the experimental output values, taking the square of this subtraction, summing the squares, and dividing the DF of the regressed model, which is $k$. Here again, the significance is given by the *p*-value, obtained directly from the IBM SPSS software. The *F*-statistic follows an *F*-distribution, which is a skewed distribution with $k$ and $n - k - 1$ degrees of freedom (DF). Since we use 1 input variable in the SLR and 2 input variables in the MLR, the number of DFs for the SLR and MLR for the *F*-distribution

are $(1, n - 2)$ and $(2, n - 3)$, where $n$ varied from 23 to 51 observations, depending on the input-output variable combination.

We will also be using another form of *F*-statistic for detecting the incremental significance obtained when adding an extra variable to the best predictor from the SLR in the MLR model. This is calculated as:

$$F_{inc} = \frac{(R^2_{MLR} - R^2_{SLR})/(k_{MLR} - k_{SLR})}{(1 - R^2_{MLR})/(n - k_{MLR} - 1)} \tag{3}$$

where $R^2_{MLR}$ and $R^2_{SLR}$ are the coefficients of determinations from the MLR and SLR models, respectively, and $k_{MLR}$ and $k_{SLR}$ are the number of input variables in the MLR and SLR models, respectively ($k_{MLR} = 2$ and $k_{SLR} = 1$), while $n$ is the number of data observations. A relative increase in the value of this *F*-statistic on adding the best predictor variable to the less-significant variable, rather than vice versa, i.e., on adding the less-significant variable to the best predictor variable, is expected. In simple terms, the increase in the $R^2$ of the model will be more pronounced when we add the best predictor variable to the less-significant variable, rather than the other way around.

### 3.3.4. Statistical Analysis Procedure: Step-by-Step

Once the correlations between the inputs and the outputs are developed, eight SLR models are built for each output; namely, GU, GD, VU, and VD, to identify the best-predicted output from the model-performance metrics. The first task was to see which output correlated highest with the inputs so that that output can be chosen to proceed with further statistical analysis, in order to isolate the most important input variable of the material or pore property of the MOFs. From this step, we also receive an initial idea as to which input variable might share the highest variance with the output; this finding is further evaluated and confirmed by developing the SLR models.

In the next step, we developed simple linear regression models between each of the 8 input variables and the outputs of the gravimetric and volumetric uptakes and deliveries, from which we observed that the prediction accuracies for gravimetric delivery (GD) and volumetric uptake (VU) outclassed those for other output variables. The Pearson's correlations were also corroborated by the highest values for the $r^2_C$ and lowest RMSEC values for GD as the output with the highest prediction accuracy, followed by VU. This is expected as we have already noted that $r^2_C$ is the square of the Pearson's correlation between the input and the output variables. Since delivery is a key parameter by which to assess methane storage performance, we chose to proceed with GD as the output to evaluate the goals of this work. In developing the SLR models, we also explored the possibility of quadratic, cubic, and exponential fits for the input variables with different outputs, before choosing to proceed with GD. For each of these fits on each of the 8 inputs, the *p*-values from the *F*-tests were also evaluated for these SLR models, with GD as the output variable on the calibration set. The *F*-test will yield a significant value for the ANOVA if the calculated value is greater than the critical value, as taken from the table corresponding to the DF of the SLR. From the *p*-values of the *t*-tests that give the significance of each regression coefficient estimate and the $r^2_C$ and RMSEC measures, the types of best fits are chosen for each input variable.

Furthermore, the order of prediction accuracy for the calibration set is also evaluated by comparing the $r^2_C$ and RMSEC values between the best fits for each input variable; it is also compared with the results from the Pearson's correlation analysis since different types of fits other than linear form were tested. The signs of the correlation coefficients for each input variable with the output GD are also noted for investigating the nature of the relationship. Now, the best fits chosen from the calibration set needed to be verified further by predicting the GD outputs using new input data from the validation set. The coefficient of determination or the percentage of variance explained in the validation set output GDs ($r^2_P$) and the root mean square errors for the prediction set (RMSEP) were evaluated; whichever property of the MOFs gave the highest $r^2_P$ and lowest RMSEP would be chosen

as the final best fit and the best predictor for the output GD. Additional confirmation was also achieved by checking for the lowest overfitting tendency of the best predictor variable and its fit-type, by evaluating the difference between the RMSEP and RMSEC values.

However, the SLR analysis alone is not sufficient to firmly determine the most important parameter because the predictions of the model using the best predictor and its type of fit, as evaluated from the validation set for the SLR models, may be strongly influenced by the other input parameters in the model due to the high inter-correlations between them, as was evident from the Pearson's correlation results, given in Section 4: Results and Discussion. In statistical terms, the inter-correlations between the input parameters in a regression model make the output variance predicted by even the best-predicting input variable, shared with other variables; thus, the shared variance between the best predictor and other input variables is not exclusive to that best-predicting variable. This phenomenon is called multicollinearity and it has been shown to pose numerous problems in regression analysis for various applications in chemical engineering [54–58]. A previous study by one of the authors of this work [59] employed rigorous statistical analysis, using similar techniques, to tetralin oxidation studies conducted in a microreactor, to determine the most important parameter that influences ketone-to-alcohol selectivity in the products of hydrocarbon oxidation. This parameter also played a role in maximizing oxygen availability in the gas-liquid system, enabling oxidation to take place. The parameter of greatest importance was the gas-liquid interfacial area; this parameter could be recommended for easier and quicker reactor design in the industry.

To tackle multicollinearity and arrive at the most important input parameter, we applied multiple linear regression (MLR) to the calibration set, which takes more than one input into consideration simultaneously to predict the output. Keeping the best predictor, along with its fit-type from the SLR models, as one variable, the MLR models were developed with all the other variables and were taken one at a time. It is obvious that when the number of input explanatory variables was increased, the prediction accuracy would also increase, irrespective of the individual correlations between that input variable and the output; hence, only 2 variables were taken at a time in the MLR models, to guard against over-estimating the coefficient of determination and creating false-high prediction accuracy. Furthermore, the standard error is directly proportional to the sum of squared errors/residuals of the model predictions and also to the correlations between the input variables themselves, while being inversely dependent on the number of samples. Hence, if the inter-correlations between the inputs increase, the standard errors in turn increase, which leads to a lower *t*-statistic, meaning that the regression coefficient term becomes less significant. This concept was used to find out which input variables were not significant compared to the best predictor from the SLR.

To further strengthen the relative importance of the best predicting input, we evaluated various statistical metrics and multicollinearity diagnostics for the MLR models by adding a variable to the best predictor variable from SLR and comparing them with those of the corresponding SLR model, looking for major changes with respect to each input parameter. The parameter or MOF property that showed the least change in these metrics and multicollinearity diagnostics, compared to other parameters, was considered the most important parameter/input variable to influence methane storage performance. These metrics and the changes that we investigated include:

(i) **Changes in signs** or a drastic decrease in the **regression coefficient estimates** from the SLR to the MLR model for an input variable make it unstable in estimating the output;

(ii) Increase in **standard errors** of the regression coefficient estimate of that variable indicates that it is rendered insignificant, compared to the other input variable;

(iii) If the **standardized coefficients** of one variable in the MLR model are higher than that of the other variable, it signifies that the variable with a higher standardized coefficient can explain the variance in the output better and will have a higher prediction accuracy, independent of the less significant variable;

(iv) **Partial correlations** of the inputs with the output indicate the influence of 1 variable on the output, in the presence of the other variables. The input variable with a higher partial correlation will be the more significant variable;

(v) **Variance inflation factors (VIFs)** are defined by $\frac{1}{1-R^2_{x_1 x_k}}$, where $R^2_{x_1 x_k}$ is the coefficient of determination when the variable $x_1$ is regressed on $x_k$, which represents the set of all other explanatory variables in the model. VIF would be 1 for simple linear regression, while a higher value in the MLR model will indicate that multicollinearity is more prevalent in that particular variable, making the other variable more significant. As with Pearson's correlation coefficient, there is the risk of a false diagnosis of multicollinearity with VIF as well, since there is no consensus on the threshold value [60]. Kutner et al. [61] suggest a minimum value of 10, while Vatcheva et al. [62] demonstrated that even a value of < 5 could be problematic. More than the absolute value, a change in VIF magnitude toward the higher side could provide evidence supporting multicollinearity, which is what is pursued in this study by comparing multiple regression models with the simple regression counterparts, as detailed in further sections in the manuscript. In addition, VIF can also be compared with $\frac{1}{1-R^2_{model}}$ to establish whether the correlation between the regressors is stronger than the overall regression model [56].

(vi) **Eigenvalues (EV) and Condition Index (CI)**
The sum of the eigenvalues of the correlation matrix (obtained through eigenvalue decomposition) will equal the number of explanatory variables in the system but the distribution of the eigenvalues across the dimensions of the matrix would point toward the presence or absence of linear dependencies [57]. If the variables are linearly independent, all eigenvalues will equal unity; in the case of correlated variables, certain dimensions would show eigenvalues that are close to 0. The latter situation indicates that the regression parameter estimates, when regressed using these input variables, would be very sensitive to changes in the data. The condition index (CI) helps in amplifying the unequal distribution of the eigenvalues and is given in Equation (1) as:

$$CI = \sqrt{\frac{\lambda_{max}}{\lambda_i}} \tag{4}$$

where $\lambda_{max}$ and $\lambda_i$ are the maximum and the $i$th eigenvalue, respectively. According to Midi et al., [58] if the $CI$ falls below 15, then multicollinearity is not a serious concern. Johnston [63] proclaimed inconsequential collinearity until $CI < 20$. Furthermore, the detection process will also be assisted by the variance decomposition proportions for each predictor, i.e., the proportion of variance for the regression coefficient estimates of each input variable that belongs to every dimension. Significantly correlated variables would have higher variance proportions, concentrated on the same eigenvalue dimension. We have considered this aspect in our study as well. Another diagnostic that has been reported in the literature but that has been used on fewer occasions is the determinant of the correlation matrix, where a lower value indicates multicollinearity. However, this diagnostic is beyond the scope of our study.

The challenge of multicollinearity that is relevant to our data is that it can render the regression coefficient estimates unstable; however, the main advantage is that it maintains them to be unbiased [54]. As highlighted in Section 3.3.1, the first step in the detection of multicollinearity is through an analysis of the pairwise product-moment Pearson correlations, which is necessary but is not a sufficient condition. Interpreting the strength of a correlation can also be subjective. Hence, other diagnostics, such as those suggested above, were utilized in this work to confirm the presence of interdependencies among the predictor variables and can be useful when proper conclusions cannot be made from the correlation matrix alone. The variable that shows the least changes in these statistical diagnostics and other measures, including the regression coefficient estimates, their standard errors, and the significance of the terms evaluated through the *t*-statistics, is the most important variable

for maximizing the methane storage performance, while the multicollinearity in the other variables is so predominant that it affects the output predictions, rendering them insignificant. In this way, we try to nullify the harmful effects of multicollinearity in our data so as to draw meaningful chemometric inferences and create tenable interpretations about the relative importance of MOF properties in predicting methane storage performance-related output variables.

This entire procedure is summarized in a **flow chart with timelines**, as shown below in Figure 2.

**Figure 2.** Step-by-step procedure for the statistical analyses followed in this work to find the most important MOF material or pore property that has the maximum influence on methane storage performance.

In conclusion, this comprehensive statistical analysis will help us to evaluate the effects of different MOF material, thermodynamic, and pore properties on methane storage performance and enable us to select the most important property with the maximum influence on the storage performance of methane, independent of other properties. The MOF design can be tuned toward minimizing or maximizing this property, according to the sign of the correlations and the other results of our study, in order to obtain maximum methane storage and delivery.

## 4. Results and Discussion

*4.1. Bivariate Pearson's Correlations*

4.1.1. Input-Output Correlations

All the calculated bivariate correlations of the 4 outputs with the 8 different inputs, along with their significance values, are given in Table 2 below.

**Table 2.** Bivariate Pearson's correlations between the various input and output variables used in this work. The value in the brackets indicates the corresponding two-tailed *p*-value of the correlation.

| Input or Output | GD [8] | GU [9] | VD [10] | VU [11] |
|:---:|:---:|:---:|:---:|:---:|
| ASA [1] | **0.96 (<0.001)** | **0.947 (<0.001)** | 0.399 (0.059) | **−0.592 (<0.001)** |
| Density | **−0.941 (<0.001)** | **−0.908 (<0.001)** | −0.406 (0.003) | **0.597 (<0.001)** |
| BET SA [2] | **0.924 (<0.001)** | **0.889 (<0.001)** | 0.298 (0.049) | −0.491 (<0.001) |
| $V_P$ [3] | **0.925 (<0.001)** | **0.883 (<0.001)** | 0.249 (0.075) | −0.558 (<0.001) |
| AV [4] | **0.918 (<0.001)** | **0.885 (<0.001)** | 0.206 (0.346) | **−0.613 (<0.001)** |
| LCD [5] | **0.819 (<0.001)** | 0.778 (<0.001) | 0.090 (0.684) | −0.501 (0.009) |
| PLD [6] | **−0.687 (<0.001)** | −0.642 (<0.001) | **−0.731 (<0.001)** | 0.246 (0.226) |
| $Q_{st}$ [7] | **−0.68 (<0.001)** | −0.374 (0.014) | −0.185 (0.24) | **0.727 (<0.001)** |

[1–8]: The expansions of the abbreviations are already provided in Section 3.1. [9]: Gravimetric Uptake; [10]: Volumetric Delivery; [11]: Volumetric Uptakes. The bold values indicate the higher values of Pearson's correlations.

From Table 2, above, it can be clearly seen that among all the output variables, GD shares the best correlation with almost all the inputs, except in the case of PLD and $Q_{st}$, for which VD and VU displayed higher correlations, respectively. These particular values, along with the corresponding *p*-values that indicate their significance, are highlighted in bold in Table 2. The number of bold entries is the highest for GD and decreases in the order GD > GU > VU > VD. Additionally, among the volumetric performance measures, VU is correlated with a higher significance to all the inputs, except with PLD. However, the correlations for the majority of the inputs with GD are much higher and are all significant (with *p*-values < 0.001), compared to the corresponding relationships with VU as the outputs. Moreover, since delivery is the ultimate goal of methane storage and capture, we proceeded with GD as the input for the rest of the statistical procedures, as also previously given in Section 3.3.4.

We can also see that among all the input parameters, ASA shared the highest correlation of 0.96 with GD, closely followed by density, which was negatively correlated with $r = -0.941$. This finding was expected because if the crystal density decreased, then the pore volume would increase, leading to a higher surface area, which was expected to translate into a higher adsorption capacity. The problem with this expectation is that a higher surface area might not always mean that all the sites are accessible, which is probably why ASA was seen to have the highest correlation with GD among all eight inputs. This is probably also the reason why crystal density ranked higher than the surface area of the MOF, as calculated by the BET method, in explaining the variance in the MOF performance measures. As discussed in the literature review, this needs further investigation as to which of the coordination or metal sites or the hydrophobic pockets show a preference for the $CH_4$ to bind with them. The bivariate correlations, in order of decreasing significances and absolute values, were pore volume, BET SA, the accessible volume (AV) parameter, LCD, PLD, and finally $Q_{st}$.

4.1.2. Input-Input Correlations

The key observation that surfaced as a result of applying Pearson's correlations to the input variables, consisting of the material, thermodynamic, and pore properties of the MOFs, is that there were linear dependencies between them. This situation is not ideal

for building a regression model with the output variables since the amount of variance explained in the GD output from one input variable ($I_1$) will be shared majorly with another input variable ($I_2$), if $I_1$ and $I_2$ are highly correlated. In this context, it was noted that density shared the highest correlation with ASA ($r = -0.99$) and the least correlation with $Q_{st}$ ($r = 0.572$); its correlations with other parameters, such as BET surface area, pore volume, and accessible volume were found to be greater than 0.89 (Table 3).

**Table 3.** Intra-input Pearson's correlations for the material, thermodynamic, and pore properties of the 83 MOFs used in this work.

| Input Variables | ASA | Density | BET SA | $V_P$ | AV | LCD | PLD | $Q_{st}$ |
|---|---|---|---|---|---|---|---|---|
| ASA | 1 | **−0.99** (**<0.001**) | **0.944** (**<0.001**) | **0.952** (**<0.001**) | **0.956** (**<0.001**) | **0.804** (**<0.001**) | −0.566 (0.003) | **−0.881** (**<0.001**) |
| Density | **−0.99** (**<0.001**) | 1 | **−0.89** (**<0.001**) | **−0.898** (**<0.001**) | **−0.917** (**<0.001**) | **−0.745** (**<0.001**) | **0.653** (**<0.001**) | 0.572 (<0.001) |
| BET SA | **0.944** (**<0.001**) | **−0.89** (**<0.001**) | 1 | **0.979** (**<0.001**) | **0.982** (**<0.001**) | **0.871** (**<0.001**) | −0.252 (0.284) | −0.49 (0.002) |
| $V_P$ | **0.952** (**<0.001**) | **−0.898** (**<0.001**) | **0.979** (**<0.001**) | 1 | **0.989** (**<0.001**) | **0.863** (**<0.001**) | −0.345 (0.084) | −0.484 (<0.001) |
| AV | **0.956** (**<0.001**) | **−0.917** (**<0.001**) | **0.982** (**<0.001**) | **0.989** (**<0.001**) | 1 | **0.902** (**<0.001**) | −0.346 (0.084) | **−0.781** (**<0.001**) |
| LCD | **0.804** (**<0.001**) | **−0.745** (**<0.001**) | **0.871** (**<0.001**) | **0.863** (**<0.001**) | **0.902** (**<0.001**) | 1 | 0.653 (<0.001) | −0.563 (0.01) |
| PLD | −0.566 (0.003) | **0.653** (**<0.001**) | −0.252 (0.284) | −0.345 (0.084) | −0.346 (0.084) | −0.237 (0.244) | 1 | **0.644** (**0.002**) |
| $Q_{st}$ | **−0.881** (**<0.001**) | 0.572 (<0.001) | −0.49 (0.002) | −0.484 (<0.001) | **−0.781** (**<0.001**) | −0.563 (0.01) | **0.644** (**0.002**) | 1 |

The bold entries in the above table (Table 3) indicate bivariate correlations that are greater than 0.6, among the various combinations of inputs that represent the material, thermodynamic, and pore properties of the MOFs used in this work. We can see that the number of bold entries is greater than the number of un-bolded ones, indicating a large presence of multicollinearity within the explanatory variables, which serve as the input variables for the regression models. As mentioned before, this phenomenon of multicollinearity reduces the accuracy of the predictions obtained from the regression models with the collinear variables as the inputs, even if the individual input-output correlations are high. It is to be kept in mind that Pearson's correlations explore the linear relationships between 2 variables; the square of this correlation value is the coefficient of determination of the SLR model between the corresponding input and output combination. Since $r^2$ gives the variance explained in the output by that input, the correlation is also an indirect measure of the explained variance in the output by that input. However, this becomes a little more complicated for a MLR model. For example, the $r^2$ for the SLR model of the linear relationship between density and the output GD is 0.886, the $r^2$ for the SLR model, considering ASA as the input and with GD as the output, is 0.921. This does not mean that 92.1% of the output variance was fully explained by ASA and the 88.6% of variance in GD was explained purely by density. This is because of the strong inter-correlation of −0.99 existing between density and ASA, and it hinders us from finding the individual contribution of these inputs to predict the output.

*4.2. Ascertaining the Prediction Accuracy and Best Predictor for GD from SLR Models*

The key contribution of this section is that it explores the nonlinear relationships between the inputs and the best-predicted output, GD, in addition to the linear model. The main difference between the SLR and the MLR models is that each input is taken one at a

time to construct the SLR models, whereas in MLR models, the best predictor from the SLR is taken, along with one other variable. We will describe the results obtained from building the SLR models on both the calibration and the validation sets in this section.

4.2.1. SLR Models Applied to the Calibration Set

Apart from the linear models, three other types of fits were investigated for establishing the best relationship between the structural properties of the MOFs and their corresponding performances. These are the quadratic, cubic, and exponential transformations of the input variable. However, caution was employed while constructing the quadratic and cubic regression models. Only the quadratic and cubic terms were considered separately in the respective regression models since adding the linear term to the quadratic and cubic models and the quadratic term to the cubic models would only increase the $r^2$; however, this does not contribute to our purpose as then we will not able to compare this model with the individual terms. For each estimate of the regression coefficient, the corresponding significance in terms of the p-value is provided and the confidence interval (CI) is calculated with the help of the *t*-statistic. If 0 happens to fall in the CI, then the coefficient estimate of that term is considered insignificant; as a result, its *p*-value and the standard error will be higher. Furthermore, the significance of the *F*-statistic for the SLR model is equal to the significance of the Pearson's correlation between the same input and output variables.

Let us analyze the SLR model with each input variable for the output GD. In the case of pore volume, the linear fit had the highest prediction accuracy, with an $r_C^2$ value of 0.855, the quadratic model being the second-best result with $r_C^2 = 0.72$. Here, the $r_C^2$ is the coefficient of determination of the calibration set. These trends were verified using the adjusted $r_c^2$ since it adds reliability and precision to the goodness of fit; the $r^2$ value alone can skew the results by considering every additional independent variable. The adjusted $r_c^2$ for these 2 fits were found to be 0.852 and 0.714, respectively. The residual standard deviation (RSD) for the linear fit, which is another way of measuring the RMSE for the calibration set (RMSEC), was less than that of the quadratic fit, thus corroborating the results shown by the $r_c^2$. Although the *F*-statistics of both the fit-types were significant (above the *F*-critical value of 4.04), the *F*-statistic value of the linear fit was more than double the value of the *F*-statistic of the quadratic fit. Furthermore, the *p*-values of the linear and quadratic terms, along with the constant terms, were all < 0.001, with low standard errors; the confidence intervals were favorable in not comprising 0 within the lower and upper bounds, due to the lower standard errors of the regression coefficient estimates of the linear and quadratic terms. These observations further confirmed that the linear fit of $V_P$ had a better prediction accuracy for GD than all the other fits explored in this work. The procedures of these calculations are described in Sections 3.3.2–3.3.4 for the readers' reference. These results from the SLR models are also in concurrence with the conclusions obtained from the Pearson's correlations regarding the relationships with the other outputs of VU, VD and GU, which suggest that the input parameters predicting GD with the best accuracy and highest significance and *F*-statistic values decrease according to the order GD > GU > VU > VD.

Next, the parameters estimated for the BET SA-GD SLR model yielded very similar results to the $V_P$-GD SLR model, where the $r_C^2$ for the former model with the linear fit was 0.854, while the value for the quadratic fit was 0.74. The similarity in the coefficients of determination for the linear fits, in the case of BET SA and $V_P$ as the input variables, can be traced back to the similarity in the Pearson's correlations of these two regressors with the output GD (Table 2). The RSD/RMSEC values for the linear and quadratic fits for this input-output combination were also found to be the same (0.015 and 0.0204 for the linear and quadratic fits, respectively). The constant terms for these two fits also turned out to be the same, both being significant, with *p*-values of <0.001. The cubic term was not considered because the $r_C^2$ was much lower (0.59) and the RMSEC was higher (0.025), although the coefficient was significant. Interestingly, the exponential fit for the BET SA-GD model yielded similar model adequacy performance measures for the $r_C^2$ as

the quadratic fit but was still less than the linear fit. Hence, the linear fit was chosen to be the best fit for BET SA from the SLR model to predict GD. At this point, due to the largely similar prediction accuracies of GD with BET SA and pore volume as the inputs and a high degree of correlation between them (Table 3), we decided to consider only one input from these two.

Which one we considered did not affect the ultimate objective of this paper, but BET SA was chosen for further study since the surface area was much more easily determinable using experimental methods than was pore volume. The works by Nematollahzadeh and Abdekhodaie [64] and Martins et al. [65] have used Wheeler's equation to determine the relationship between the pore diameter, total pore volume, and the surface area as $diameter = \frac{4V}{S_{BET}}$, but the situation is not as straightforward in real adsorbents. The relationship between the pore volume and the surface area entirely depends on the shape of the pores and the distribution of active sites inside the adsorbent. It is conventional to expect that the surface area for adsorption should increase if the pore volume increases, which is also relayed by Wheeler's equation mentioned above and can be realized geometrically as well. Consider a non-porous spherical adsorbent with a radius $R$; its surface area will be $4\pi R^2$ and the pore volume will be 0. If we make a cylindrical pore with radius $r$ and a depth $l$ in the spherical adsorbent, then the new surface area available for adsorption will increase to $2\pi rl + 4\pi R^2$ and the pore volume will be $\pi r^2 l$. The creation of more pores will only increase the pore volume and the surface area available for adsorption, but in real adsorbents, there have been cases observed where the BET surface area decreased with the increase in pore volume; however, this is a rarity and might be due to experimental error as well [65]. Additionally, the fewer the number of highly inter-correlated variables as inputs to our regression models, the better; thus, we omit the pore volume and consider only the BET surface area as one of the inputs in this work.

The next variable we considered is density, for which the best two fits were exponential and linear, as inferred from their highest $r_C^2$ values of 0.909 and 0.886 for the fits, respectively. Since this was an SLR model with a single explanatory variable, the adjusted $r_C^2$ values were not much different from the actual $r_C^2$ values, differing only by 0.002 units. The exponential fit for the density variable performed better than both the BET SA and pore volume in predicting the GD for the calibration set; this finding aligns with one of the objectives of the paper and also the hypotheses of this work (refer to the works discussed in Section 2.3) [30]. The RMSEC values aligned with those of the $r_C^2$ values, and both the exponential and linear terms were highly significant. The quadratic fit was the third-best fit for the density-GD SLR model.

The two best fits for each input variable, along with their corresponding $r_C^2$, $F$-statistic values, the $p$-values of the constant, the first terms in the SLR models, and the signs of the bivariate correlations are given in Table 4. Figure 3 is shown along with Table 4 in order to easily visualize the trends. Among all the input variables, the linear fit of ASA displayed the highest $r_C^2$ and lowest RMSEC values for the prediction of GD in the calibration set. However, it is worth noting that the F-statistic (that is, the ratio between the average of the variances of the regression model and the average of the variances of the prediction errors) for the exponential fit of density was double (490) that of the linear fit of ASA (243), indicating that although the explained variance in GD was higher with the linear fit of ASA as the input, the overall significance of the model was greater when density was used as the input, with the coefficient of determination in the calibration set for the exponential fit of density being lower by only 0.01 units (or 1% explained variance). In fact, the $F$-statistic value of the linear fit of ASA is of the same magnitude as that of the linear fit from BET SA with GD, although the $r_C^2$ of BET SA was lower. One more point in favor of the significance of density that was observed was that the $p$-value of the constant term for the linear fit of ASA was 0.047 (Table 4), which still falls within the 5% limit for the term to be significant, whereas the constant terms for both the exponential and linear fits for density were < 0.001, which meant that they were more statistically significant than the best fit for ASA. All these values are summarized in Table 4.

**Table 4.** The best fits of the six selected input variables depicting the material, pore, and thermodynamic properties from the SLR models.

| Output | Input | Top 2 Best Performing Fits | $r_C^2$ | *F*-Statistic | *p*-Value for $b_0$ * | *p*-Value for $b_1$ ** | Best Fit according to the Calibration Set | Sign of the Correlation |
|---|---|---|---|---|---|---|---|---|
| GD | ASA | Linear | 0.92 | 243 | 0.047 | <0.001 | Linear | Positive |
| | | Quadratic | 0.88 | 151 | <0.001 | <0.001 | | |
| GD | Density | Exponential | 0.91 | 490 | <0.001 | <0.001 | Exponential | Negative |
| | | Linear | 0.89 | 381 | <0.001 | <0.001 | | |
| GD | BET SA | Linear | 0.85 | 240.5 | <0.001 | <0.001 | Linear | Positive |
| | | Quadratic | 0.74 | 116 | <0.001 | <0.001 | | |
| GD | AV | Linear | 0.84 | 113 | 0.011 | <0.001 | Linear | Positive |
| | | Exponential | 0.74 | 60 | <0.001 | <0.001 | | |
| GD | LCD | Linear | 0.67 | 42 | 0.184 | <0.001 | Quadratic | Positive |
| | | Quadratic | 0.66 | 41 | <0.001 | <0.001 | | |
| GD | $Q_{st}$ | Cubic | 0.54 | 46 | <0.001 | <0.001 | Cubic | Negative |
| | | Exponential | 0.52 | 42 | <0.001 | <0.001 | | |

\* $b_0$ is the constant term in the SLR model; ** $b_1$ is the coefficient term that corresponds to the transformation of the input variable.

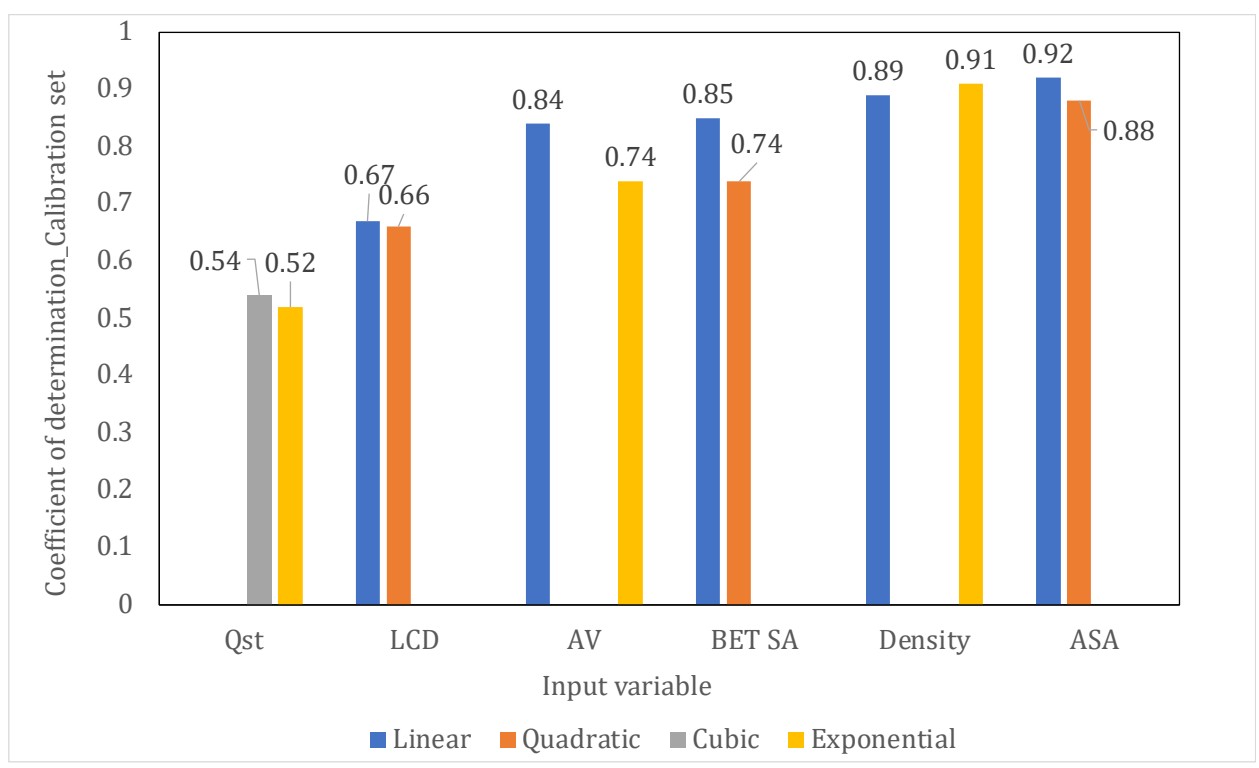

**Figure 3.** Bar chart representing the data for $r_C^2$ in Table 4, above: the top two best-performing fits for each input regressor, with GD as the output.

All these observations imply that choosing the best explanatory variable that predicts the output GD with the least error and highest accuracy is not as simple as checking for the highest coefficient of determination and the lowest RMSE; other factors, such as the significance of the overall model and the individual coefficient regression estimates, need to be considered as well. However, all the fits that were found to have the highest

prediction accuracy for GD that were tested need to be tested on the validation set to completely confirm with high confidence that they are indeed the best fits with the least overfitting tendency.

The linear fit of accessible volume ($r_C^2 = 0.84$, RMSEC = 0.018) predicts the GD just as well as BET SA ($r_C^2 = 0.85$, RMSEC = 0.015) but exhibits less than half the significance of the overall model (*F*-statistic = 113) obtained for the BET SA-GD input-output combination (*F*-statistic = 240), as can be seen in Table 4 and Figure 3. The significance of the overall model for the exponential fit of AV, which was the second-best fit for the AV-GD relationship, was almost half the significance of the linear fit, although it had the same $r_C^2$ as that of the quadratic fit for BET SA, which was the second-best fit for BET SA in predicting the output GD. We attempted to make physical sense of why the significance of the accessible volume parameter was lower than accessible surface area, crystal density, and BET surface area in predicting GD. A large accessible volume might not necessarily mean that the number of active sites where the $CH_4$ adsorption can take place will also increase. It means only that the methane molecules can access these pores; the presence of active sites for adsorption is not guaranteed. As explained in the previous discussion in this section on how an increase in the pore volume might increase the surface area that is available for adsorption, this additional surface area created might not necessarily contain a large number of active sites; even if the active sites are present, they might not be reachable by the methane molecules due to their location deep inside or due to the pore size being smaller than the size of the methane molecules.

The next variable that we analyzed was the LCD. The highest $r_C^2$ values for the LCD-GD relationship occurred for both the linear and the quadratic fit as 0.67 and 0.66, respectively. The *F*-statistic values were also so close (42 and 41, respectively) that it was difficult to choose the best fit from the calibration set. However, the quadratic fit for LCD was chosen since the constant term of the linear fit was not significant at the 95% confidence level (the *p*-value was 0.187), while the *p*-value for the constant term in the quadratic fit was <0.001, making it an easy choice for consideration as the best fit to be tested in the validation set. It should be noted that the *p*-values of the linear and quadratic terms for the LCD-GD SLR model were both <0.001, as determined by the IBM SPSS software. On the other hand, the PLD, which is closely related to the LCD, was found to be the least effective for explaining the variance in GD, with $r_C^2$ values of 0.47 and 0.51 for the cubic and exponential fits, respectively, which values were higher than the linear and quadratic fits. Although the coefficient of regression estimates for these terms, along with their constants, were significant, with *p*-values < 0.001, the *F*-statistic values were the least (18 and 21, respectively) among all the input-GD SLR models, implying that PLD was not effective in predicting the output GD. It is useful to understand the difference between LCD and PLD with respect to the adsorbent geometry and how the two influence its bulk properties. In the literature, researchers define LCD as the largest sphere that can be inserted in the porous material without overlapping with any of the atoms, while the PLD is defined as the largest sphere that can freely diffuse through the porous network without overlapping with any of the atoms [66–68]. Hence, there is not much to differentiate between these two structural properties that describe a porous network and that also determine the efficiency of the interconnected pores in affecting the adsorbent performance, in terms of methane uptakes and deliveries. Hung et al. [67] also describe PLD as the aperture size of the material and showed that PLD can be more effective in increasing the selectivity of MOFs toward membrane gas separations. Since our work does not utilize membranes, and the model performance of the SLR for predicting the GD was better with LCD as the input, we omitted PLD from our further analysis with the validation set and MLR models.

The isosteric adsorption energy had lower $r_C^2$ values than that for LCD, as can be seen in Table 4 and Figure 3. However, it is interesting to note that the overall significance of the model for the best-fitting cubic and exponential transformations of $Q_{st}$ were higher than that of LCD, making the LCD-GD SLR the least significant model overall for both its best fits (Table 4). It is also worth noting that the only explanatory variable for which the cubic

transformation was the best fit was the $Q_{st}$, but the thermodynamic descriptors are thought to have more importance in affecting the rotational and translational motions of methane during its capture through MOFs; this falls under the topic of mechanisms, which was beyond the scope of this work. This finding was evident in our co-author's previous work, where thermodynamic properties such as the entropy of adsorption were considered, and optimum isosteric heats were calculated by the use of different empirical correlations [35]. Thus, the six explanatory variables that describe the structural descriptors of the MOFs with one thermodynamic property were considered to have been tested in the validation set for confirming the best predictor/regressor/input/explanatory variable, along with its linear/nonlinear transformation from the SLR model that predicted GD the best.

### 4.2.2. SLR Models Applied to the Validation Set

The coefficient of regression estimates obtained from the SLR models that were constructed based on the calibration set are used to predict the GD with the corresponding new input variables values for the selected six regressors that are identified in Section 4.2.1 and given in Table 5. The best fits obtained from the SLR models on the calibration sets are used for this purpose; model performance measures, such as the coefficient of determination of prediction ($r_P^2$), residual standard deviations on the validation set (RMSEP), and the overfitting tendency indicated by the absolute value of the difference between the RMSEP and RMSEC are provided in Table 5. Figure 4 provides the plot for a variation of $r_P^2$ for the best fit for each input variable with the output GD for easier visualization.

**Table 5.** Model performance measures for the best-fitting SLR models, developed using the calibration set and tested on the validation set. The values are in decreasing order of $r_P^2$ .

| Best-Fit of the Input | $r_P^2$ | RMSEP | $r_C^2$ | RMSEC | abs (RMSEP—RMSEC) |
|:---:|:---:|:---:|:---:|:---:|:---:|
| Density (Exponential) | 0.88 | 0.015 | 0.91 | 0.011 | 0.004 |
| BET (Linear) | 0.85 | 0.02 | 0.85 | 0.015 | 0.012 |
| ASA (Linear) | 0.76 | 0.021 | 0.92 | 0.013 | 0.008 |
| AV (Linear) | 0.75 | 0.031 | 0.84 | 0.018 | 0.013 |
| LCD (Quadratic) | 0.36 | 0.053 | 0.66 | 0.027 | 0.026 |
| $Q_{st}$ (Cubic) | 0.26 | 0.046 | 0.54 | 0.027 | 0.019 |

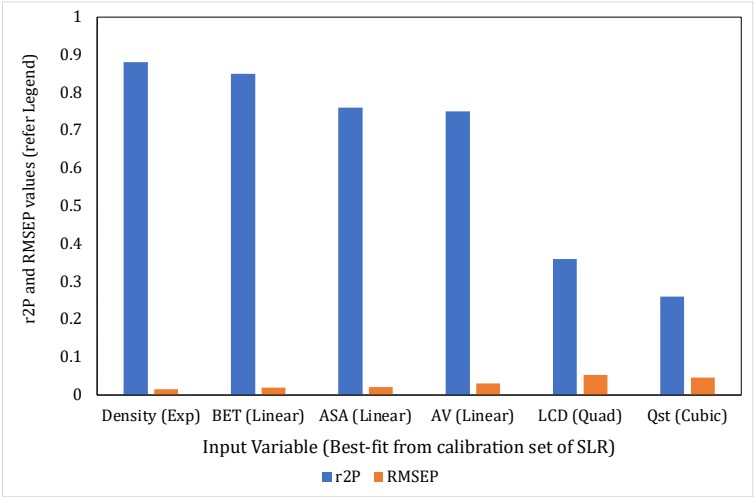

**Figure 4.** Comparison of $r_P^2$ and RMSEP values for the different best-fit transformations of the input variables of the SLR models from the calibration set tested on the validation set.

It can clearly be seen in Figure 4 that the best-performing input from the calibration set (ASA) was overshadowed by the exponential fit of the density, with a much higher $r_P^2$ and a much lower RMSEP (0.015). The RMSEP for the linear fit of ASA was 0.021. In fact, even the linear fit of density was found to have a higher $r_P^2 = 0.81$ and a lower RMSEP of 0.018. BET SA also performed better than ASA when the best-fitting linear forms of the input variables were compared for the prediction of GD on the validation set, with $r_P^2 = 0.85$ and an RMSEP value of 0.02. The linear fits of density and the quadratic fits of BET SA were found to have $r_P^2$ values of 0.81 and 0.76, respectively, and, when compared to all the fits of ASA, were shown to outperform them in RMSEP as well. Another aspect that plays a key role in establishing density as the best-performing input variable in predicting GD with a test set is the lowest overfitting tendency displayed. This is seen from the lowest value (0.004) of the difference between the RMSEP and the RMSEC, as seen in Table 5. This tells us that when taking density as the input variable, the GD can be predicted at high levels of accuracy for any random test data, with the constants of regression estimates obtained from this calibration set. This is important both for the design of the MOF and also to eliminate any offline measuring instruments to determine the values of the deliveries and uptakes of methane.

Some other interesting observations from Table 5 and Figure 4 are that the $r_P^2$ values tend to be lower than the $r_C^2$ values for the majority of the SLR models. Only the BET SA-GD model displayed equal values for both the coefficients of determination of the calibration and validation sets. This is because the validation set consists of new "unseen" input values that the SLR model takes; it tried to predict the output variable with the regression coefficient estimates obtained from the calibration set, which consists of a limited number of observations. Of course, the $r_P^2$ will be higher when the number of observations increases in the calibration set, which enables the regression coefficients to be estimated closer to the population values. Another observation was that the best fits of the LCD (quadratic) and $Q_{st}$ (cubic) input variables displayed the least prediction accuracy for GD in the test set, as was the case in the calibration set, with the greatest tendencies for overfitting as well. Overall, from the SLR models, we can conclude that density was the best-performing input property of the MOF that predicted the GD with the highest prediction accuracy, tested over a single validation set. Next, the multicollinearity between the input variables needs to be addressed, to check whether the maximum variance explained in the output for the test sets was indeed exclusive to density alone.

### 4.3. Assessing the Interdependencies between the Input Variables Using the MLR Models

In order to compare the coefficients, standard errors, and other parameters mentioned in Sections 3.3.3 and 3.3.4 for the MLR models with those for the SLR models, the MLR models were also constructed with the data points from the calibration set. The MLR models were constructed with two variables as the input and GD as the output. One variable among the two input variables was density since it was observed to be the best performing regressor from the SLR models alone; each of the other input variables was considered one by one along with density, to build five different MLR models. The points of comparison for the performance parameters between the MLR and SLR models are summarized in Tables 6 and 7 and Figures 5 and 6. Table 6a–c give a comparison of the regression coefficient estimates, their standard errors, and the standardized coefficients (model considered without the intercepts—please refer to Section 3.3.2), respectively. Table 7a–d provide the model significance parameters, such as the *t*-statistic values, the confidence intervals, partial correlations, their changes from the zero-order bivariate correlations for each coefficient of regression estimates, and the *F*-statistic values of the SLR and MLR models, respectively. Lastly, Figures 5 and 6 show the eigenvalues (EVs) for each dimension, the corresponding condition indices, and the proportions of variance decompositions on each dimension for each term in the MLR model. The SLR models are named S1, S2, S3, S4, S5, and S6, while the MLR models are named M1, M2, M3, M4, and M5 for convenience of referencing. All

the values of the performance parameters are only provided for the input variable term in the SLR and MLR models; the constant term values are not given.

**Table 6.** (**a**) Comparison of the regression coefficient estimates for the two-input MLR models, with density as one of the inputs and the individual SLR models. The details of the MLR models are given in the footnote below the table. (**b**) Comparison of the standard errors (SE) of the regression coefficient estimates for the two-input MLR models, with density as one of the inputs and the individual SLR models. The details of the MLR models are given in the footnote below the table. (**c**) Comparison of the standardized coefficients for the two-input MLR models, with density as one of the inputs and the individual SLR models. The details of the MLR models are given in the footnote below the table.

| (a) | | | | | | |
|---|---|---|---|---|---|---|
| **Input Variable** | **Coefficient Estimates for SLR Models** | **Coefficient Estimates for MLR Models** | | | | |
| | | **M1** | **M2** | **M3** | **M4** | **M5** |
| Density_exp [1] | −1.49 | −0.32 | −0.31 | −0.44 | −0.34 | −0.38 |
| BET SA_linear [2] | $3.62 \times 10^{-5}$ | $4.8 \times 10^{-6}$ | | | | |
| ASA_linear [3] | $3.83 \times 10^{-5}$ | | $5.8 \times 10^{-6}$ | | | |
| AV_linear [4] | 0.092 | | | −0.02 | | |
| LCD_quad [5] | $3.2 \times 10^{-4}$ | | | | $2.3 \times 10^{-5}$ | |
| Q_st_cubic [6] | $-1.3 \times 10^{-5}$ | | | | | $-1 \times 10^{-6}$ |

| (b) | | | | | | |
|---|---|---|---|---|---|---|
| **Input Variable** | **SEs for SLR Models** | **Standard Errors (SEs) of Regression Coefficient for MLR Models** | | | | |
| | | **M1** | **M2** | **M3** | **M4** | **M5** |
| Density_exp [1] | 0.016 | 0.058 | 0.018 | 0.085 | 0.038 | 0.027 |
| BET SA_linear [2] | $2 \times 10^{-6}$ | $6 \times 10^{-5}$ | | | | |
| ASA_linear [3] | $2 \times 10^{-6}$ | | $1.9 \times 10^{-5}$ | | | |
| AV_linear [4] | 0.009 | | | 0.023 | | |
| LCD_quad [5] | $5 \times 10^{-5}$ | | | | $4 \times 10^{-4}$ | |
| Q_st_cubic [6] | $2 \times 10^{-6}$ | | | | | $1.4 \times 10^{-4}$ |

| (c) | | | | | | |
|---|---|---|---|---|---|---|
| **Input Variable** | **Standardized Coefficients for SLR Models** | **Standardized Coefficients for MLR Models** | | | | |
| | | **M1** | **M2** | **M3** | **M4** | **M5** |
| Density_exp [1] | 0.94 | 0.84 | 0.82 | 1.19 | 0.92 | 1.03 |
| BET SA_linear [2] | 0.92 | 0.12 | | | | |
| ASA_linear [3] | 0.96 | | 0.15 | | | |
| AV_linear [4] | 0.92 | | | 0.23 | | |
| LCD_quad [5] | 0.81 | | | | 0.06 | |
| Q_st_cubic [6] | 0.73 | | | | | 0.08 |

[1]—exponential best fit from SLR model S1; [2]—linear best fit from SLR model S2; [3]—linear best fit from SLR model S3; [4]—linear best fit from SLR model S4; [5]—quadratic best fit from SLR model S5; [6]—cubic best fit from SLR model S6. M1 = Density_exp + BET_linear; M2 = Density_exp + ASA_linear; M3 = Density_exp + AV_linear; M4 = Density_exp + LCD_quad; M5 = Density_exp + Q_st_cubic.

**Table 7.** (**a**) Comparison of the *t*-statistic values of the coefficient regression estimates for the two-input MLR models, with density as one of the inputs and the individual SLR models. The details of the MLR models are given in the footnote below the table. (**b**) Comparison of the confidence intervals of the coefficient regression estimates for the two-input MLR models, with density as one of the inputs and the individual SLR models. The details of the MLR models are given in the footnote below the table. (**c**) Comparison of the partial correlations of the coefficient regression estimates for the two-input MLR models, with density as one of the inputs and the individual SLR models. The details of the MLR models are given in the footnote below the table. (**d**) Comparison of the *F*-statistic values for the 2-input MLR models with density as one of the inputs and the individual SLR models. The details of the MLR models are given in the footnote below the table.

**(a)**

| Input Variable | *t*-Statistics for SLR Models | *t*-Statistic Values for MLR Models | | | | |
|---|---|---|---|---|---|---|
| | | **M1** | **M2** | **M3** | **M4** | **M5** |
| Density_exp [1] | −23 | −5.6 | −1.7 | −5.2 | −8.9 | −14.3 |
| BET SA_linear [2] | 15.5 | 0.8 | | | | |
| ASA_linear [3] | 15.6 | | 0.3 | | | |
| AV_linear [4] | 10.6 | | | −1 | | |
| LCD_quad [5] | 6.4 | | | | 0.5 | |
| Q$_{st}$_cubic [6] | −6.6 | | | | | 1.1 |

**(b)**

| Input Variable | Confidence Intervals for SLR Models | | Confidence Intervals for MLR Models | | | | | | | | | |
|---|---|---|---|---|---|---|---|---|---|---|---|---|
| | Lower Bound (LB) | Upper Bound (UB) | **M1** | | **M2** | | **M3** | | **M4** | | **M5** | |
| | | | LB | UB | LB | UB | LB | UB | LB | UB | LB | UB |
| Density_exp [1] | −3 | −0.39 | −2.1 | −0.44 | −3.1 | −0.7 | −2.6 | −0.6 | −2.1 | −0.5 | −3.3 | −0.4 |
| BET SA_linear [2] | $3 \times 10^{-5}$ | $4 \times 10^{-5}$ | $-7 \times 10^{-6}$ | $2 \times 10^{-5}$ | | | | | | | | |
| ASA_linear [3] | $3 \times 10^{-5}$ | $4 \times 10^{-5}$ | | | $-4 \times 10^{-5}$ | $5 \times 10^{-5}$ | | | | | | |
| AV_linear [4] | 0.074 | 0.11 | | | | | −0.07 | 0.02 | | | | |
| LCD_quad [5] | $2.2 \times 10^{-4}$ | $4.2 \times 10^{-4}$ | | | | | | | $-6 \times 10^{-5}$ | $1 \times 10^{-4}$ | | |
| Q$_{st}$_cubic [6] | $-2 \times 10^{-5}$ | $-9 \times 10^{-6}$ | | | | | | | | | $-1 \times 10^{-6}$ | $4 \times 10^{-6}$ |

**(c)**

| Input Variable | Partial Correlations for SLR Models | Partial Correlations for MLR Models | | | | |
|---|---|---|---|---|---|---|
| | | **M1** | **M2** | **M3** | **M4** | **M5** |
| Density_exp [1] | −0.94 | −0.66 | −0.45 | −0.76 | −0.9 | −0.92 |
| BET SA_linear [2] | 0.92 | 0.13 | | | | |
| ASA_linear [3] | 0.96 | | 0.07 | | | |
| AV_linear [4] | 0.92 | | | −0.22 | | |
| LCD_quad [5] | 0.81 | | | | 0.13 | |
| Q$_{st}$_cubic [6] | −0.73 | | | | | 0.18 |

**Table 7.** *Cont.*

| | | (d) | | | | |
|---|---|---|---|---|---|---|
| | | **F-Statistic Values for MLR Models** | | | | |
| **Input Variable** | **F-Statistics for SLR Models** | **M1** | **M2** | **M3** | **M4** | **M5** |
| Density_exp [1] | 490 | −0.66 | −0.45 | −0.76 | −0.9 | −0.92 |
| BET SA_linear [2] | 241 | 0.13 | | | | |
| ASA_linear [3] | 243 | | 0.07 | | | |
| AV_linear [4] | 0.92 | | | −0.22 | | |
| LCD_quad [5] | 0.81 | | | | 0.13 | |
| Q$_{st}$_cubic [6] | −0.73 | | | | | 0.18 |

[1]—exponential best fit from SLR model S1; [2]—linear best fit from SLR model S2; [3]—linear best fit from SLR model S3; [4]—linear best fit from SLR model S4; [5]—quadratic best fit from SLR model S5; [6]—cubic best fit from SLR model S6. M1 = Density_exp + BET_linear; M2 = Density_exp + ASA_linear; M3 = Density_exp + AV_linear; M4 = Density_exp + LCD_quad; M5 = Density_exp + Q$_{st}$_cubic.

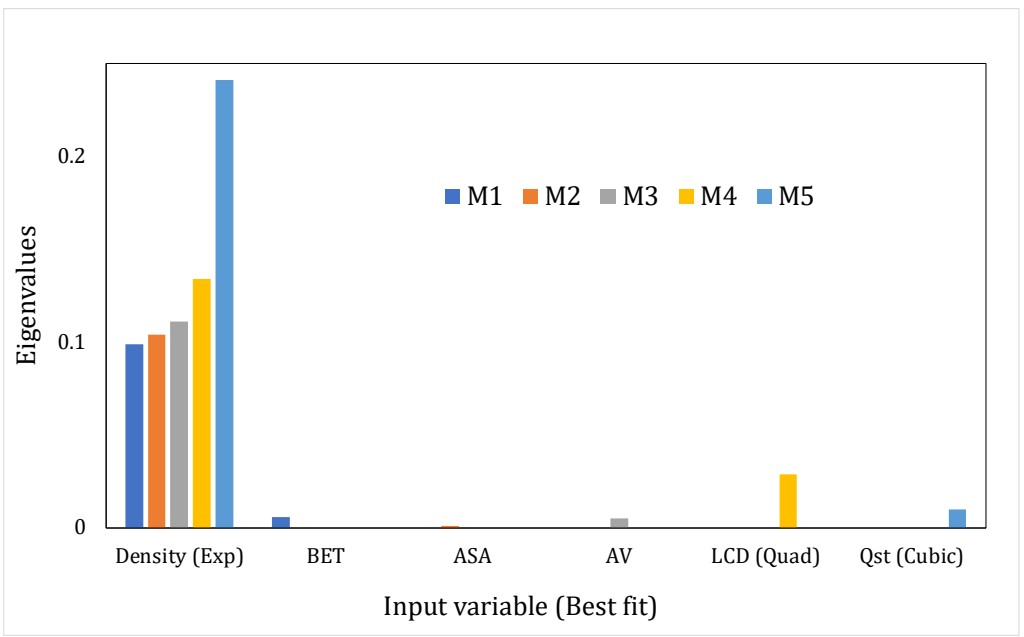

**Figure 5.** Eigenvalues for the 2 dimensions in 5 MLR models corresponding to the best-fit model of each input variable that was obtained from the SLR analysis.

The dependent and independent variables are both standardized before constructing the regression model to eliminate the effects of the units of the different variables involved in the unstandardized regression coefficients. The standardization process is performed by converting each variable into its z-score by subtracting the mean of the observations for that variable from each original data point and dividing it by the standard deviation of the variable, considering all the data points. The standardized coefficients are interpreted, such that a change of one standard deviation in the input is associated with a change in the $\beta$ number of standard deviations in the output. This implies that the higher the standardized coefficient, the higher the effect of that input variable on the dependent variable; we looked for changes in the coefficients in the MLR model as compared to the corresponding SLR model for the individual variables, as given in Table 6c. For simplification purposes, absolute values of the standardized coefficients are considered. It is to be noted that the standardized coefficients for the SLR models are the same as the Pearson's bivariate correlations between the best-fit transformation of that regressor with the output GD. This is given in the second column of Table 6c. Next, we can see that the standardized coefficients in the MLR models drastically decrease for all other variables except for density. For BET

SA input, the coefficient decreased 7.5 times from 0.92 in the SLR model to 0.12 in the MLR model M1, while for the linear form of ASA, the standardized coefficient decreased ~6.5-fold to 0.15 for the MLR model from 0.96 in the SLR model. Similarly, the standard coefficients for the remaining inputs, such as the linear form of AV, the quadratic form of LCD, and the cubic form of $Q_{st}$, decreased by ~4 times, 13.5 times, and ~9 times, respectively. However, the coefficients for density displayed minimal changes for all the MLR models, compared to their SLR counterpart, and it was interesting to note that the values actually increased for models M3 and M5, as can be seen in Table 6c. This tells us that the impact of the density variable was the highest in contributing to the variance change in GD when combined with all the other input variables considered in this study.

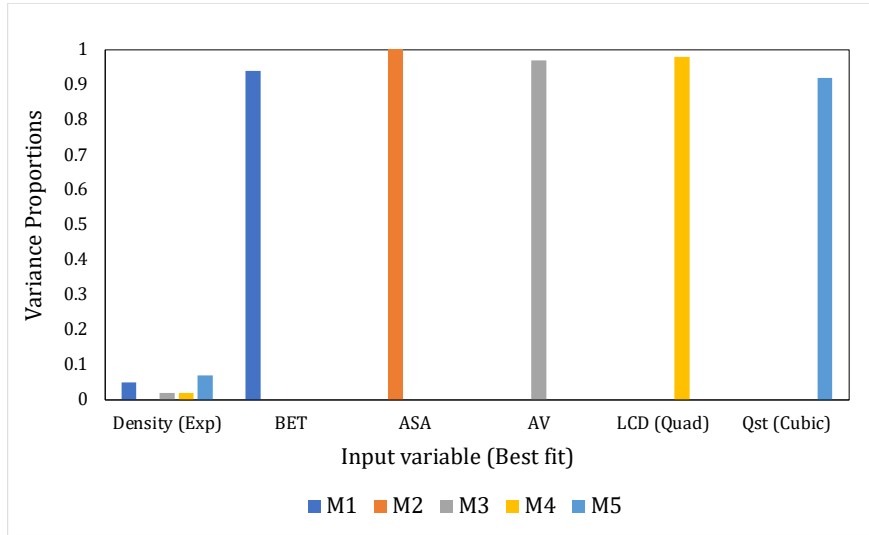

**Figure 6.** Variance proportion distribution corresponding to the second term in the five MLR models, corresponding to the best-fitting model of each input variable with GD that was obtained from the SLR analysis.

The standard errors given in Table 6b provide a similar trend to that shown by the standardized coefficients, where the best fit from the SLR model for density was seen to have the least increase in SE, when combined with the other variables for predicting the GD in the MLR models. It was seen that the SE increased by at least an order of magnitude for all the other input variables: for the linear forms of BET SA and ASA (both from $10^{-6}$ to $10^{-5}$), for the linear form of AV from 0.009 to 0.023, for the quadratic transformation of LCD from $10^{-5}$ to $10^{-4}$, and lastly, for the cubic transformation of $Q_{st}$ by two orders of magnitude from $10^{-6}$ to $10^{-4}$. The increase in standard errors of the coefficient regression estimates also brought down their *t*-statistic values, as seen in Table 7a, since the *t*-statistic is inversely dependent on the SE (Equation (2)). Furthermore, with regard to the values of the regression coefficient estimates themselves, there was a decrease in the order of magnitude for BET SA, ASA, LCD, and $Q_{st}$, while there was a change in sign from +ve in the SLR to -ve in the MLR model M3, involving AV as the input. A decrease in magnitude as well as a change in sign from the SLR to MLR model signified that the contribution of that input variable had decreased and become insignificant compared to density, for which there was no change in sign; in fact, the weight of the negative value of the coefficient regression estimate increased but remained at the same order of magnitude.

Table 7a–c provide the *t*-statistic values, the confidence intervals, and the partial correlations of the SLR and MLR models for each coefficient regression estimate while Table 7d provides the *F*-statistic values for the SLR and MLR models. It can be seen that there was a substantial decrease in the *t*-statistic values of all five input variables except density, from the SLR models to the corresponding MLR models (Table 7a). For example, the *t*-statistic for BET SA and ASA decreased from 15.5 to 0.8 and 0.3, respectively,

decreasing from their individual SLR models to those in M1 and M2 models when they were combined with density. These implied decreases of 19- and 52-fold for these two variables, respectively, while those for the density input decreased only by 4 times and 10 times from its SLR models to M1 and M2, respectively (Table 7a). The larger decrease in the *t*-statistic value of density in model M2 with ASA as compared to other models was due to the increased correlation between them, but the decrease in the significance of the ASA coefficient term was much higher as it decreased 52-fold. Moreover, the absolute values of the *t*-statistic were always much higher than the best fits of other variables, such as BET SA, ASA, AV, LCD, and $Q_{st}$ in models M1 to M5 by 7, 6, 5, 16, and ~13 times, respectively. The other main inference to be drawn from Table 7a is that the decrease in the *t*-statistic values of the best fits of AV (10 times), LCD (11-fold), and $Q_{st}$ (~6-fold) from their SLR models was much higher than the decrease in those of the density for the corresponding M3 (~4-fold), M4 (~2.5 times), and M5 (~1.5 times) models. This signifies clearly that the output GD was much more sensitive to the coefficient regression estimates of density than it was to the other regressors.

As can be seen in Table 7b, none of the confidence intervals of density in either of the SLR or MLR models include 0 in them. This means that they are all statistically significant in having an influence in the output that is unlike the other explanatory variables, which have a negative lower bound and a positive upper bound. The inclusion of 0 in the confidence interval implies that the coefficient regression estimate for that term can be 0, which means that there will be a minimal contribution of that regressor for a change in the output. The partial correlations are the same as the standardized coefficients for the SLR models and are equal to the bivariate Pearson's correlations between that input variable and the output GD. However, the partial correlations of density with the output for the MLR remain almost the same when combined with LCD_quad and $Q_{st}$_cubic, as can be seen in Table 7c. The corresponding partial correlations decreased steeply to 0.13 from 0.81 for LCD and displayed a change in sign for $Q_{st}$, with a fourfold decrease from an absolute value of 0.73 for the SLR model to 0.18 for the MLR model, along with displaying a change in sign. Similarly, the influence of the other three input variables on the output decreased sevenfold, ~13.5 times and 4.5 times for BET SA, ASA, and AV, which can be attributed to the decrease in their partial correlations when combined with density for the MLR models. On the contrary, the minimum value to which the partial correlation for the density decreased was for the M2 model, where it was combined with the linear fit of ASA (which had the highest $r_C^2$ in the SLR model), with a 50% decrease to $-0.45$ in the MLR model from $-0.94$ in the SLR model. For M1 and M3, the partial correlations of density were higher than those of BET SA and AV by five times and ~four times, respectively, signifying that the maximum degree of influence among the input variables on GD was density.

Moving on, it was very interesting to note that all the MLR models were significant in terms of the overall variances of the models as compared to their mean squared errors, as indicated by the *F*-statistic values in Table 7d. This is important because it was observed that the *F*-statistic values of the M3, M4, and M5 models increased significantly, and this can be attributed to the high *F*-statistic value of 490 for the density-GD model S1. However, the F-statistic values of the MLR model M1 decreased by half from that of S1 and was close to that of S2, while those for M2 decreased by ~3.5 times from that of S1 and were almost half of S3. These can be attributed to the negative effects of the input variables of BET SA and ASA on the combined MLR model, while density had tremendous positive effects when combined with AV, LCD, and $Q_{st}$, resulting in an increase in the overall significance of models M3, M4, and M5 by 1.5, 4, and 6 times, respectively (Table 7d).

The variance inflation factors (VIFs) for the MLR models were all found to increase compared to the SLR models, for which the VIF is always 1. The maximum increase was shown by M2 and the minimum increase was seen to be for M5; this was purely a consequence of the inter-correlations between the density and the other input variables. The presence of multicollinearity was confirmed with the calculation of VIFs for the MLR models. Next, the eigenvalues for each of the 2 dimensions corresponding to both explanatory

variables, with density as one regressor in the MLR models, are provided in Figure 5. It is worth noting that the sum of the eigenvalues across the dimensions equals the number of explanatory variables involved in the model, plus the constant term, i.e., 2 for SLR and 3 for MLR, with 2 input variables, as in this work. It can be seen in Figure 5 that the EVs for all other input variables except density were two orders of magnitude lower than that of density in the same MLR model. This implied that the regression coefficient estimates are very sensitive to small data changes in that variable and that this renders that variable insignificant. This also corresponded to the maximum value of the condition index given in brackets in Figure 5. Thus, the EVs also establish that density was the most important variable among all the explanatory variables in predicting MOF performance.

The variance proportions were concentrated on the third dimension for all the MLR models, as seen in Figure 6, where the third dimension corresponds to all the other input variables apart from density. This implied that these input variables were insignificant as compared to density in influencing the output It can also be said that these input variables do not share the true explained variance in the output due to their high degree of inter-correlations, whereas density clearly does. Therefore, from all of these results and the accompanying discussion, it can be proclaimed that the density (which is a material property) of the MOFs is the most important property in predicting gravimetric delivery and affect their storage and delivery performances to the highest degree, as compared to the other pore and thermodynamic properties.

## 5. Future Extensions of Our Work

Since we developed the models at 35 bar, it will also be worthwhile to see if the model is applicable at other pressures, including 65 bar, subject to having access to sufficient experimental data points. Working at different operating conditions will only change the uptakes and deliveries along with the adsorption energies, keeping the pore parameters and structural descriptors the same. This will be handled well by our versatile and robust statistical data-based model. Another by-product of our study may be to predict performance in the methane capture process, where the output performance metrics are the same but the best predictor among the MOF properties might be different. The hypothesis, in that case, may be that the isosteric heat of adsorption for the $CH_4$-MOF system might be the most important property that influences the methane capture process. Nonlinear relationships are expected in methane capture modeling because thermodynamic data are involved.

As a final comment, this model can also be extended to covalent organic frameworks (COFs) since the nature of the input and output data will be the same; this model will be expected to have a high prediction accuracy for the COF-methane system as well. The performances of 2 new COFs synthesized in the work by Mendoza-Cortes et al. [69] were seen to be comparatively similar to those of MOFs; the addition of vinyl bridges to the structure improved the performance. At present, the inputs to our statistical SLR and MLR models will be the structural descriptors of the COFs, such as surface area, pore volume, density, and the isosteric heats of adsorption; in addition, we can also use the lattice parameters obtained from molecular dynamic simulations as inputs in order to see whether they will be relatively important compared to other input variables and, hence, whether or not they can be used for design purposes. Therefore, the process will be similar; we can affirm that this kind of comprehensive statistical model has not previously been applied to COFs adsorbing methane in the literature. This will be the direction of our future work, but we can either perform the experiments ourselves or obtain the data from the literature. Working on experimental data is most important to ensure the real-time applicability of our models.

## 6. Conclusions

The two main goals of this work were: (i) to investigate the prediction accuracies of methane uptakes and deliveries in MOF adsorbents, as experimentally determined at

298 K and 35 bar, by employing rigorous statistical data analysis tools; (ii) to identify the most important design property of the MOF, among a set of structural, pore, and thermodynamic descriptors, that was able to cause maximum changes to the performance measures of the MOF, affecting methane storage and delivery, in order to aid in reducing the sector's global emissions and contribute toward net zero emissions in the near future. Both these objectives have successfully been accomplished; it was seen in this work that density, a material property of the MOF, predicted the MOF performance measures for methane storage and delivery with the highest accuracy and lowest errors, compared to other pore and thermodynamic properties of the MOF. This was established by carrying out a rigorous statistical treatment of the experimental data, consisting of 83 observations and 8 properties of the MOF, which served as the input, and 4 MOF performance parameters that served as the output to the regression models. Simple linear regression (SLR) and bivariate Pearson's correlations fulfilled the first objective, whereby density proved to be the best regressor for predicting the output MOF performance metrics, with the regression model being constructed based on a calibration set and tested on a new validation set. The model proved to be versatile and robust when the multicollinearity existing between the various input variables was tackled using multiple linear regression (MLR) by constructing the models with density as one of the input variables and each of the other regressors included one by one to form five MLRs with two input variables each. On comparing the regression coefficient estimates, their standard errors, partial correlations of each input, standardized coefficients, and the *t*-statistic values, it was seen that the density showed minimal changes for these statistical metrics when the MLR models were compared with their SLR counterparts. Additionally, multicollinearity diagnostics, such as VIFs, comparison of eigenvalues for each input dimension, condition indices, and the distribution of variance proportions across the three dimensions for the MLR models, all confirmed with full confidence that density was the most statistically significant variable in predicting the output, both as a stand-alone parameter and when considered along with other MOF properties as well. Thus, we conclude that density is more significant than surface area, as determined by the BET method, as well as accessible surface area when altering the MOF adsorption performance for methane.

**Author Contributions:** Conceptualization, E.M.; methodology, K.S.; software, K.S.; validation, K.S. and E.M.; formal analysis, K.S.; investigation, K.S.; resources, E.M.; data curation, E.M.; writing—original draft preparation, K.S.; writing—review and editing, K.S. and E.M.; visualization, K.S. and E.M.; supervision, E.M.; project administration, E.M.; funding acquisition, K.S. All authors have read and agreed to the published version of the manuscript.

**Funding:** This research was funded by Startup Grant Fund at UAE University with grant number 12N104.

**Institutional Review Board Statement:** Not applicable.

**Informed Consent Statement:** Not applicable.

**Data Availability Statement:** The data are contained within the article.

**Acknowledgments:** The authors would like to thank and acknowledge the UAEU Startup Research Grant for the funding provided towards this project.

**Conflicts of Interest:** The authors declare no competing financial interest.

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
