# Peer review of "Development of a High-Accuracy Statistical Model to Identify the Key Parameter for Methane Adsorption in Metal-Organic Frameworks"

_analytica, doi:10.3390/analytica3030024_

Round 1

Reviewer 1 Report

This manuscript by Sivaramakrishnan et al. develops a high-accuracy statistical model to identify the key parameter for methane adsorption in metal-organic frameworks. The authors claim that that density is more significant than surface area determined by the BET method and accessible surface area as well, in altering the MOF adsorption performance for methane. This is an interesting study, I recommend a minor revision.

1. I suggest authors to add a scheme describing the main content of this study at the begining of the manuscript.

2. Authors just provided some tables in their manuscript, can authors provide some additional figures for clarity.

3. Can this statistical model be applied to other framework materials, such as covalent organic frameworks (COFs)?

Author Response

Dear reviewer,

We have attached all the responses to your comments and suggestions in the below word document and made the required changes to our manuscript. Kindly request you to please see the attachment. Thank you so much for your time and detailed consideration in reviewing our manuscript and making it better than before. 

Yours sincerely,

Kaushik and Eyas. 

Author Response

Dear Reviewer, 

We sincerely thank you very much for the detailed review that you have provided with much efforts. We have addressed all your points and included the details in the attached document as well as corrected them in the manuscript. We have also marked the changes in the manuscript as comments for your kind reference. Addressing your comments have made our manuscript much better than the previous version and we tank you for recommending to publish our work.

Yours sincerely,

Kaushik and Eyas.

Reviewer 3 Report

This contribution from the group of Sivaramakrishnan describes a high-accuracy statistical model for methane adsorption in metal-organic frameworks. In their study, they used the simple and multiple linear regression (SLR and MLR) combined with different types of multicollinearity diagnostics, partial correlations, standardized coefficients, changes in regression coefficient estimates and their standard errors applied. It is quicker, provides a deeper insight into experimental data and has not been previously employed in the methane storage literature. As the result, they identify the most important design property of the MOF among a set of structural, pore and thermodynamic descriptors. I think the paper will be of interest to the readership of analytica and I recommend the paper is accepted with some minor changes. Some specific comments are shown below:

1.          Can this model be used for any MOF if they have different functional groups?

2.          “(i) to investigate the prediction accuracies of methane uptakes and deliveries in MOF adsorbents experimentally determined at 298 K and 35 bar, by employing rigorous statistical data analysis tools”

What happens at different temperature and pressure?

Author Response

Dear Reviewer, 

We sincerely thank you very much for the detailed review that you have provided with much efforts. Your positive response was a real boost for us. We have addressed both your points and the responses are as follows:

  • Yes, this model can be used for any MOF with different functional groups since the nature of the input and output data will remain the same and that is what matters most for our statistical model. Moreover, the fact that we have considered more than 80 MOFs that have a broad spread of central metals with various complex ligands has made our model very versatile, flexible and robust. Hence, it is suitable for any new data set that arises from modifying the functional groups of MOFs
  • 298 K and 35 bar were used as the experimental condition because for methane storage applications like for natural gas-powered vehicles, the upper operating pressure used for storage is 35 bar and 298 K. In addition, the United States (US) Department of Energy’s (DOE) program for CH4 storage systems called the methane opportunities for vehicular energy (MOVE) program set the storage targets at a pressure of 3.5 MPa (35 bar) and 298 K. This is why we used this pressure. The conclusions from this study are applicable at higher pressures like 298 K and 65 bar and 298 K and 100 bar. It was previously discovered that trends obtained from methane storage correlations at 298 K and 35 bar like the correlation between gravimetric storage capacity and BET surface area can also be applied at higher pressures such as 298 K and 65 bar and 298 K and 100 bar (Reference: Quantitative Structure-Property Relationships from Experiments for CH4 Storage and Delivery by Metal-Organic Frameworks). At different temperature and pressure, the isosteric adsorption energies (one of the inputs) and the uptakes and deliveries (outputs) might be different but our model is versatile enough to deal with this difference and has been thoroughly built so as to give good prediction accuracies at different operating adsorption conditions. Changing the temperature and pressure will not really affect other input parameters such as the density, pore surface area, pore volume, LCD, PLD, accessible surface area and volume.

We have also indicated these comments in our manuscript. Addressing your comments have made our manuscript much better than the previous version and we tank you for recommending to publish our work.

Yours sincerely,

Kaushik and Eyas